# Selective ensembles in supported palladium sulfide nanoparticles for alkyne semi-hydrogenation

Davide Albani[1], Masoud Shahrokhi[2], Zupeng Chen[1], Sharon Mitchell[1], Roland Hauert [3], Núria López [2] & Javier Pérez-Ramírez[1]

Ensemble control has been intensively pursued for decades to identify sustainable alternatives to the Lindlar catalyst (PdPb/CaCO$_3$) applied for the partial hydrogenation of alkynes in industrial organic synthesis. Although the geometric and electronic requirements are known, a literature survey illustrates the difficulty of transferring this knowledge into an efficient and robust catalyst. Here, we report a simple treatment of palladium nanoparticles supported on graphitic carbon nitride with aqueous sodium sulfide, which directs the formation of a nanostructured Pd$_3$S phase with controlled crystallographic orientation, exhibiting unparalleled performance in the semi-hydrogenation of alkynes in the liquid phase. The exceptional behavior is linked to the multifunctional role of sulfur. Apart from defining a structure integrating spatially-isolated palladium trimers, the active ensembles, the modifier imparts a bifunctional mechanism and weak binding of the organic intermediates. Similar metal trimers are also identified in Pd$_4$S, evidencing the pervasiveness of these selective ensembles in supported palladium sulfides.

[1] Institute for Chemical and Bioengineering, Department of Chemistry and Applied Biosciences, ETH Zürich, Vladimir-Prelog-Weg 1, 8093 Zürich, Switzerland. [2] Institute of Chemical Research of Catalonia (ICIQ), and The Barcelona Institute of Technology Av., Països Catalans 16, 43007 Tarragona, Spain. [3] EMPA, Swiss Federal Laboratories for Materials Science and Technology, Uberlandstrasse129, 8600 Dübendorf, Switzerland. These authors contributed equally: Davide Albani, Masoud Shahrokhi. Correspondence and requests for materials should be addressed to N.Lóp. (email: nlopez@iciq.es) or to J.Pér-Rír. (email: jpr@chem.ethz.ch)

The ultimate mission of researchers working in heterogeneous catalysis is the identification of active ensembles with a high density to enable efficient turnover while avoiding undesired intermolecular interactions, and exhibiting long-term stability to ensure a well-defined performance. Nevertheless, the fulfillment of all these requirements in one catalytic material has yet to be accomplished, even for well-established reactions like heterogeneously-catalyzed hydrogenations studied since the 1880's by Wilde, Sabatier, and Senderens[1,2].

The chemo- and stereoselective hydrogenation of functionalized alkynes is a key catalytic transformation applied industrially at different scales, ranging from the gas-phase multi-ton purification of olefin streams for polymer manufacture[3] to the smaller three-phase batch preparation of high-added-value fine chemicals[4,5]. Examination of the rich scientific literature emphasizes the wide dominance of palladium-based catalysts, due to the ability to dissociate hydrogen, and the importance of preventing the undesired hydrogenation of competing functional groups through electronic poisoning with suitable modifiers (Fig. 1)[6–35]. Similarly, oligomerization paths (C–C couplings) must be suppressed by the strict control over the ensemble size. Thus, the synthesis of trimeric ensembles[36–39], and the presence of electronic modifiers reducing the hydrogen uptake of Pd and the formation of subsurface hydrides, are key parameters for achieving high selectivity both in gas- and three-phase operations.

Since 1952, the manufacture of building blocks for pharmaceuticals, vitamins, agrochemicals, and fragrances via alkyne semi-hydrogenation[4,40,41] has been conducted over the well-known lead-poisoned palladium (5 wt.%) supported on calcium carbonate (Lindlar catalyst), despite drawbacks of poor metal utilization and toxicity[42,43]. However, the recent disruptive legislations restricting the use of hazardous compounds have steered the innovation and creativity of scientists towards the development of sustainable alternatives. This has enabled the incorporation of new families of hydrogenation catalysts based on: (i) alternative metals[1,2,7,9,10,30], (ii) alloy and intermetallic compounds[8,11–16,18–21], (iii) ligand-modified nanoparticles[6,17,22,25], (iv) transition metal oxides[26,27,34,35], and (v) supported single atoms[24,31,32]. While these share better ensemble definition than unmodified Pd, intrinsic drawbacks like instability against high pressure induced segregation for intermetallics[44–46], accessibility constraints in ligand-modified and single atom catalysts[31,47], and poor $H_2$ activation on oxides[48] have been identified.

In the search for new catalysts, very few works have studied the modification with p-block elements[22,28,29,49]. In particular, sulfur-modified Pd catalysts could be envisaged mimicking the strategy of ensemble and electronic density control in enzymes. In fact, transition metal sulfides are applied for hydrotreating in oil refineries due to their resistance to poisons and ability toward $H_2$ activation[50,51], and have also been studied for electrochemical hydrogen evolution[52]. Recently, a crystalline $Pd_4S$ phase was shown to exhibit promising performance in the high pressure gas-phase hydrogenation of simple alkynes impurities in petroleum-derived olefin streams, although no molecular-level understanding was gathered[33,53,54]. Since the Pd-S phase diagram shows full mixing in all compositions and a continuum of closely related stable phases[55], it offers room to tailor the active phase with the most selective ensemble and amount of electronic poisoning at the atomic scale.

By introducing a facile and scalable procedure to prepare crystal phase and orientation controlled $Pd_3S$ nanoparticles supported on carbon nitride, this article demonstrates that sulfur can play a multifunctional role enabling unparalleled hydrogenation performance in the continuous-flow three-phase semi-hydrogenation of linear, internal, and bulky functionalized

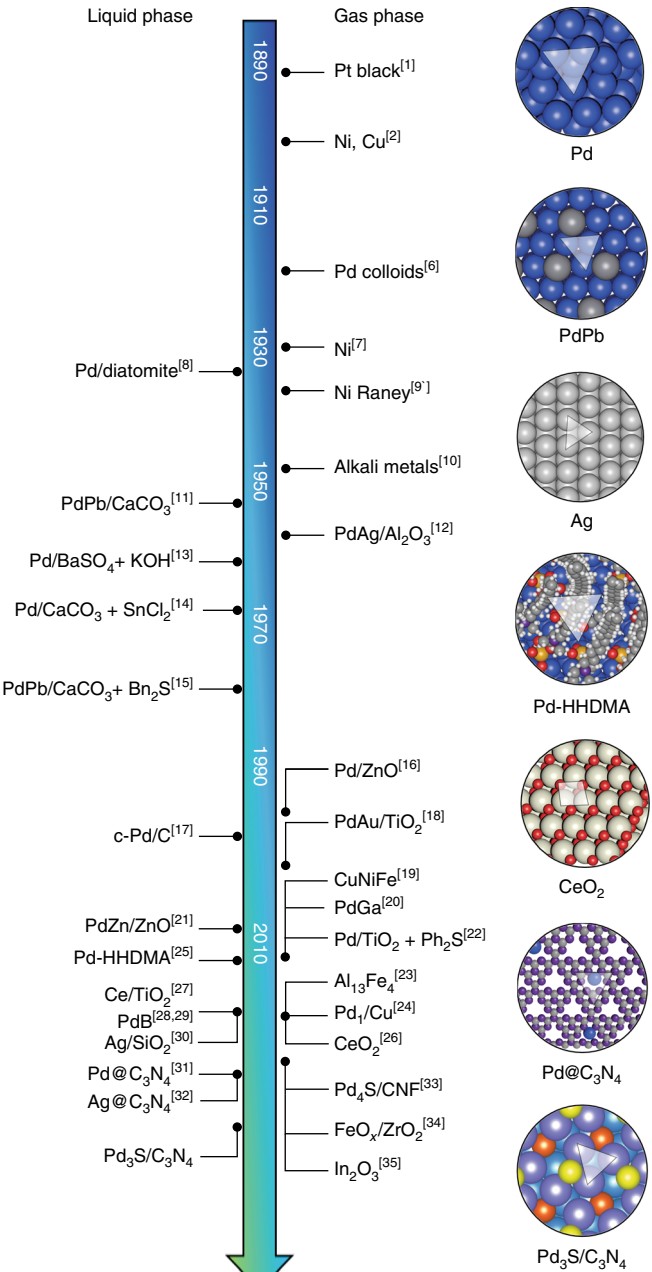

**Fig. 1** Survey of alkyne semi-hydrogenation catalysts. Chronological development for gas- and liquid-phase applications tracing from the pioneering works of Wilde[1] and Sabatier[2] to the more recent discoveries[6–35]. Six prominent catalyst families can be distinguished based on (i) supported metal nanoparticles, (ii) intermetallics, alloys, and Lindlar-type formulations where Pd is poisoned by a second metal, (iii) ligand-modified or hybrid materials featuring an organic surfactant that partially covers the surface of Pd nanoparticles, (iv) metal oxides, (v) supported single atoms, and (vi) Pd modified by p-block elements. The current industrially applied catalysts are indicated in bold. Surface structures of representative examples of each catalyst family are illustrated, highlighting the catalytic ensemble

alkynes. This is linked to the presence of trimeric metal ensembles, which are also shown to be present in palladium sulfides of different stoichiometry by the preparation and detailed interrogation of the supported $Pd_4S$ phase. The superiority with respect to other major families of hydrogenation catalysts is

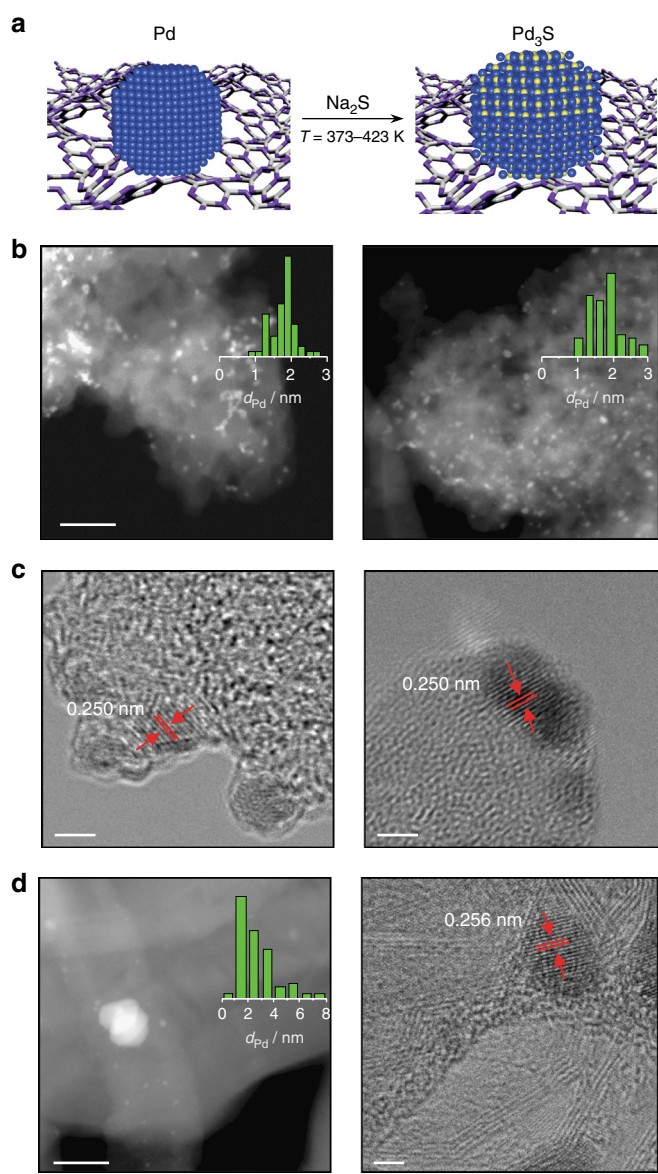

**Table 1 Properties of the catalysts studied**

| Catalyst | Pd[a]/wt.% | $S_{BET}$[b]/ m² g⁻¹ | $V_{pore}$[c]/ cm³ g⁻¹ | $d_{Pd}$[d]/nm | $D_{CO}$[e]/% |
|---|---|---|---|---|---|
| Pd/C₃N₄ | 0.4 | 278 | 0.74 | 1.7 | 63 |
| Pd₃S/C₃N₄-423[f] | 0.4(1.9)[g] | 265 | 0.51 | 1.9 | 34 |
| Pd₄S/CNF | 1.0 | 28 | 0.09 | 2.5 | 36 |
| PdPb/CaCO₃ | 4.5 | 10 | 0.03 | 15 | 6 |
| Pd-HHDMA | 0.5 | 170 | 0.64 | 8.0 | 37 |
| Pd@C₃N₄ | 0.5 | 155 | 0.26 | 0.4 | — |

[a]ICP-OES
[b]BET method
[c]Volume of $N_2$ adsorbed at $p/p_0 = 0.95$
[d]TEM
[e]CO chemisorption
[f]Temperature of sulfidation in K
[g]Molar Pd/S ratio, the sulfur content was determined by elemental analysis

**Fig. 2** Synthetic approach and microscopy images of the palladium sulfide catalysts. **a** Graphical depiction of the sulfidation of palladium nanoparticles supported on graphitic carbon nitride. The average particle size increased from 1.2 to 1.8 nm upon treatment with Na₂S (see Supplementary Note 1), consistent with the distortion of the palladium lattice expected upon introduction of sulfur atoms. Scale bars: 30 nm. **b** HAADF-STEM image with the particle size distribution (inset) of Pd₃S/C₃N₄-423 fresh (left) and after 50 h on stream (right) excluding the occurrence of sintering. Scale bars: 2 nm. **c** HRTEM image of Pd₃S/C₃N₄-423 fresh (left) and after 50 h on stream (right) confirming the preserved crystalline structure. **d** HAADF-STEM image (left) and the derived Pd particle size distribution (inset), and HRTEM (right) image of Pd₄S/CNF. Scale bars: 30 and 2 nm, respectively

established by determining the hierarchy of key descriptors for the performance.

## Results

**Synthesis of supported palladium sulfides.** Sulfided palladium phases ($Pd_xS_y$) can be readily prepared by the post-synthetic modification of supported metal nanoparticles with sulfur-containing compounds such as $H_2S$, $Na_2S$, $PdSO_4$, and Pd thiolates. Several works have examined the phase evolution on treatment of Pd/C with $H_2S$[56], which often leads to mixed crystal phases[57], but this toxic and highly corrosive gas is synthetically unattractive. Here, aqueous sodium sulfide was applied as an alternative, less widely studied sulfidation agent and the phase selectivity was monitored at different temperatures (Fig. 2a). Mesoporous graphitic carbon nitrite ($C_3N_4$) was targeted as a carrier due to its well-known ability to stabilize palladium with high dispersion, including as single atoms (Pd@C₃N₄)[31], which was expected to facilitate the sulfidation treatment. Interestingly, despite exhibiting high stability in oxidative and reductive environments at elevated temperatures, tests in the temperature range 373–423 K over Pd@C₃N₄ revealed that isolated palladium atoms were not stable against sulfidation treatments, transforming into nanoparticles, as evidenced by scanning transmission electron microscopy (STEM) (Supplementary Fig. 1a). For this reason, palladium was introduced via the wet deposition of palladium chloride coupled with an additional sodium borohydride reduction step to promote the formation of palladium nanoparticles (Pd/C₃N₄) with an average diameter of 1–2 nm and a very narrow size distribution.

Analysis of the bulk properties of Pd/C₃N₄ and the sulfided analogs (Pd₃S/C₃N₄-T, where Pd₃S was the only crystalline phase identified and $T = 373–473$ K) confirmed the uptake of sulfur, absence of metal leaching, the negligible incorporation of sodium (< 2 ppm), and structural integrity of the carrier upon sulfidation treatment (Table 1). The latter was verified by equivalent treatment of the metal-free C₃N₄ establishing the preserved crystallinity and negligible sulfur uptake (Supplementary Fig. 2). Consistent with the low content and high dispersion of palladium, no metal-containing phases were detected by X-ray diffraction (XRD) in any of the catalysts supported on C₃N₄. To determine whether the treatment resulted in the formation of one or more $Pd_xS_y$ phases or sulfur doping localized at the surface, high-resolution transmission electron microscopy (HRTEM) analyses were undertaken. Examination of the HRTEM images revealed the formation of crystalline Pd₃S nanoparticles, characterized by an interplanar distance of 0.25 nm corresponding to the (202) facet[56], in all of the sulfided samples on C₃N₄ (Fig. 2c and Supplementary Fig. 3). Elemental mapping by energy-dispersive X-ray spectroscopy (EDX) further confirmed the close proximity of palladium and sulfur and absence of any sulfidation of the carrier (Supplementary Fig. 4).

To gain insight into the thermodynamically preferred surface terminations of Pd₃S, the stability of several stoichiometric low-index planes was considered by density functional theory (DFT). The simulations indicate that Pd₃S has two low-energy crystal facets: (001) and (202), both exhibiting inequivalent palladium and sulfur atoms (Fig. 3 and Supplementary Fig. 14). The Pd₃S

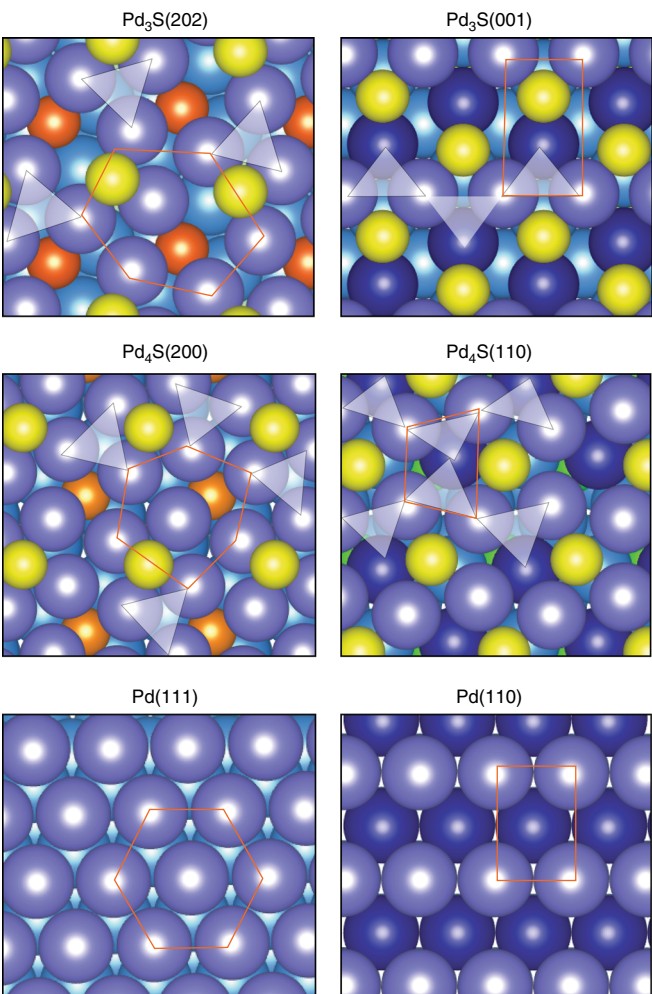

**Fig. 3** Structural resemblance of Pd$_3$S and Pd$_4$S to pristine Pd surfaces. Top views of the low-energy terminations. The structural correspondence of the sulfide surfaces with the most closely related palladium surface Pd(111) or Pd(110) is highlighted with an orange hexagon or rectangle. The Pd atoms on the Pd$_3$S and Pd$_4$S surfaces belonging to the ensemble are connected with a white triangle. Inequivalent palladium and sulfur atoms are identified with different colors (for more details see Supplementary Note 3). Color code: Pd(1)$_{surf}$ (purple), Pd(2)$_{surf}$ (blue), Pd$_{bulk}$ (light blue), S(1)$_{surf}$ (yellow), S(2)$_{surf}$ (orange)

(001) surface displays Pd atoms arranged in chains and surrounded by S atoms, where rectangular patterns resembling those of the (110) surface of an fcc structure are found between the surface Pd neighbors. On the Pd$_3$S(202) surface, each Pd is surrounded by four Pd atoms and one S. Therefore, both surfaces show triangular Pd ensembles with an area of 7.09 Å$^2$ separated by S atoms. While on Pd$_3$S(202), the triangular ensembles belong to the topmost layer, on Pd$_3$S(001) they are formed by two surface and one subsurface Pd atoms. The surface energy ($\gamma_s$) of these terminations differs by 0.15 J m$^{-2}$ with the (001) facet slightly favored (Supplementary Table 4).

Comparative inspection of the Pd$_3$S(202) and Pd(111) surfaces sheds light on the observed preferential development of the former, identifying their close geometrical relationship, maintaining the same pseudo-hexagonal arrangement of the surface Pd (111) atoms in the fcc structure typically adopted by this metal. In contrast, the Pd$_3$S(001) surface displays a closer resemblance to the rectangular pattern of Pd(110). Therefore, starting from the metallic Pd nanoparticles the mild sulfidation treatment can be

seen as a lattice distortion to incorporate S anions, leading to a kinetically-controlled process where the formation of the (202) surface implies the lowest number of distortions. This agrees with earlier results identifying the preferential development of Pd$_3$S (202) surface terminations upon sulfidation of carbon-supported palladium nanoparticles with hydrogen sulfide[56].

The chemical state of Pd at the surface of the sulfided catalysts was probed by X-ray photoelectron spectroscopy (XPS) (Supplementary Fig. 5). All the samples display Pd $3d_{5/3}$ peaks at 335.5 and 337.9 eV assigned to Pd$^0$ and Pd$^{2+}$ species, respectively[53]. Evaluation of the spectra shows that the relative ratio of Pd$^0$/Pd$^{2+}$ increases upon sulfidation, pointing to the formation of sulfided Pd species with an average oxidation state close to 0. Note that there is still a debate on the XPS signals attributed to Pd$_x$S$_y$ species; some works reporting the presence of cationic Pd species[58] and other observing a 'metallic' nature. The latter were attributed to the possible formation of a solid solution between Pd and S. To address this point and discern the bonding character, the difference charge density of Pd$_3$S crystal and the charge transfer from Pd to S have been evaluated by the Bader charge analysis (Supplementary Note 3). The results point to a high degree of covalency, with a polar Pd–S bond and negligible electron transfer, which is in agreement with the experimentally observed tendency towards Pd$^0$. In addition to charge effects, XPS shifts could also arise due to the finite size of the small nanoparticles in the catalyst that can also lead to deviations from the bulk values. To further explore the effect of sulfur poisoning on the catalysts electronic properties, projected $d$-band densities of states were calculated for all of the systems studied (Supplementary Fig. 15). Consistent with the XPS observations, down-shifts to –1.80 and –1.90 eV are evidenced for Pd$_3$S(202) and Pd$_3$S(001), respectively, compared to the reference value of –1.39 eV for Pd(111).

For reference purposes, a supported Pd$_4$S catalyst was prepared via thermal decomposition in H$_2$ flow of PdSO$_4$ on carbon nanofibers (CNF)[33]. This procedure is less attractive since for every mole of Pd$_4$S produced it evolves three moles of toxic gaseous SO$_2$ or even H$_2$S. Furthermore, as previously reported, it lacks of a precise control over the particle size distribution (1–20 nm). Characterization by XRD confirms the formation of the Pd$_4$S phase (Supplementary Fig. 2a), and contrarily to Pd$_3$S/C$_3$N$_4$, the resulting crystals mainly expose the lowest-energy (200) facet (Fig. 2d). Interestingly, a similar theoretical analysis of the Pd$_4$S (200) surface reveals that it also features Pd trimers spaced by S atoms as well as a small fraction of Pd-square arrangements (Fig. 3). Consistent with the lower amount of sulfur, analysis by XPS evidences the closely metallic nature of surface palladium (peak at 335.5 eV, Supplementary Fig. 5) and a downshift of the Pd $d$-band to −1.70 eV, a slightly less poisoned state than Pd$_3$S.

**Performance in alkyne semi-hydrogenation**. The sulfided palladium catalysts were evaluated in the continuous-flow three-phase semi-hydrogenation of 2-methyl-3-butyn-2-ol, an important building block in the synthesis of vitamin E, at various temperatures and pressures (Fig. 4a, b and Supplementary Fig. 7) and benchmarked against state-of-the-art materials (Lindlar-type PdPb/CaCO$_3$, ligand-modified Pd-HHDMA, Pd@C$_3$N$_4$ the properties of which are provided in Table 1 and Supplementary Table 1). Compared to the Lindlar catalyst, which achieves ~90% selectivity to 2-methyl-3-buten-2-ol and the remaining 10% to 2-methyl-2-butanol, Pd$_3$S/C$_3$N$_4$-423 is fully selective to the alkene and shows almost a 3.5-fold higher reaction rate (0.3 and 1.0 × 10$^3$ h$^{-1}$, respectively, at 303 K and 1 bar). Under the same conditions, Pd-HHDMA and Pd@C$_3$N$_4$ catalysts are slightly less active (0.6 and 0.2 × 10$^3$ h$^{-1}$, respectively) although they display

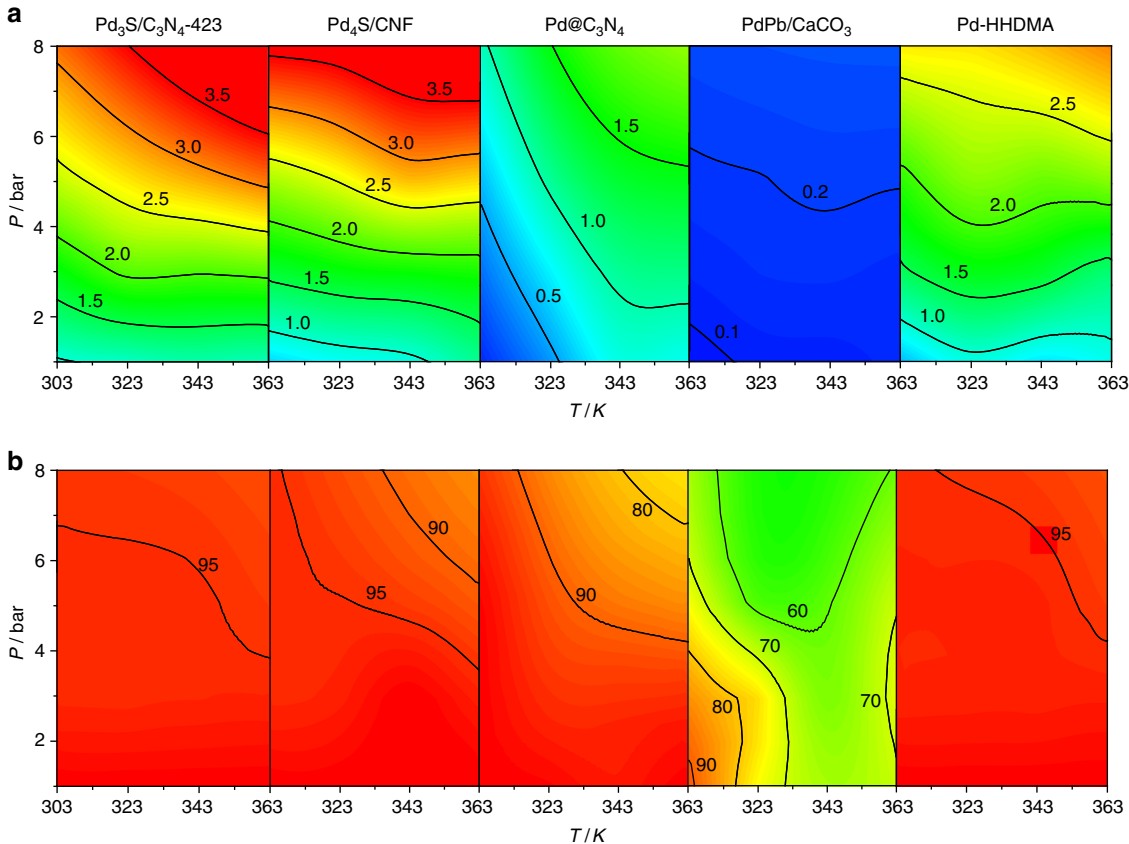

**Fig. 4** Evaluation of the alkyne semi-hydrogenation performance of the catalysts. **a** The contour plots map the effect of temperature and pressure on the rate (in $10^3$ h$^{-1}$) and **b** selectivity to 2-methyl-3-buten-2-ol (in %) of Pd$_3$S/C$_3$N$_4$-423 compared to Pd$_4$S/CNF, PdPb/CaCO$_3$, Pd-HHDMA, and Pd@C$_3$N$_4$. Conditions: $W_{cat} = 0.1$ g, $T = 303$ K, $F_L$(alkyne + toluene) $= 1$ cm$^3$ min$^{-1}$, $F_G$(H$_2$) $= 36$ cm$^3$ min$^{-1}$

excellent selectivity (100%). Pd$_4$S/CNF displays similar selectivity to Pd$_3$S/C$_3$N$_4$-423 under all conditions applied, but with slightly inferior reaction rates, for instance at 303 K and 1 bar the rate is 30% lower (0.7 compared to $1.0 \times 10^3$ h$^{-1}$ than on Pd$_3$S/C$_3$N$_4$-423). Interestingly, Pd/C$_3$N$_4$ presents lower selectivity to the alkene (90%), but similar activity to Pd$_3$S/C$_3$N$_4$-423, likely due to the higher dispersion of palladium in the untreated sample (Table 1).

To provide a broader comparison among all state-of-the-art systems and extrapolate these conclusions, the reaction temperature and pressure were varied to study the influence on the rate and selectivity in the semi-hydrogenation of 1-hexyne, a widely studied model alkyne, where the reaction can be accompanied by the *cis-trans* isomerization of the alkene product (Supplementary Fig. 8). The contour maps clearly show the superiority of using *p*-block elements for modifying Pd-based catalysts. Note that the only side product formed over the palladium sulfides was *n*-hexane (<5%). Unlike in the cases of Pd-HHDMA, PdPb/CaCO$_3$, and Pd@C$_3$N$_4$, no traces of isomers like *cis* and *trans* 2-hexene were produced on Pd$_x$S catalysts. Furthermore, the similarly high selectivity observed below 343 K and 5 bar, points to the resistance of Pd$_3$S/C$_3$N$_4$-423 towards the formation of β-hydrides. Consistently, analysis by the temperature-programmed reduction in H$_2$ did not evidence any peak due to β-hydrides[5]. The DFT results indicate that the adsorption of H atoms in subsurface positions is endothermic by 0.14 eV. Since it was not possible to reach higher alkyne conversions in our continuous-flow set up, additional batch tests were conducted to evaluate the selective performance under more challenging conditions. An impressive 95% selectivity to 2-methyl-3-buten-2-ol was preserved even at

80% conversion over the Pd$_3$S and Pd$_4$S phases, whereas much more significant losses were evidenced for the PdPb/CaCO$_3$ (80%) and Pd-HHDMA (85%) catalysts.

To gain insight into the comparative reaction mechanisms over Pd and Pd$_x$S surfaces, DFT calculations were performed (details in Supplementary Note 4). Acetylene was chosen as a model alkyne to obtain robust answers to the challenging reaction networks. The validity of using this surrogate was verified by the parallel adsorption and reaction profiles observed for acetylene and 2-methyl-3-butyn-2-ol over the Pd$_3$S catalysts (Supplementary Fig. 17). Although the adsorption of 2-methyl-3-butyn-2-ol is slightly more exothermic, due to the hydrogen bond between the alcohol group of the alkynol and the surface sulfur atom, this interaction does not change the reaction profile as is present for all intermediates and products. Also, under liquid-phase operations the reaction environment has a limited effect on the state of the catalyst[59]. Alkyne hydrogenation follows a Horiuti-Polanyi mechanism[59,60] (Fig. 5, Supplementary Tables 5–7). The first step of the reaction is the dissociation of molecular hydrogen that occurs homolytically at surface Pd sites over Pd(111), Pd$_4$S(110), Pd$_4$S(200), and Pd$_3$S(001), while the alternative heterolytic path[61] features low energy only on the Pd$_3$S(202) surface. This results in the storage of hydrogen atoms in the form of thiol groups (SH) on the Pd$_3$S(202) surface. C$_2$H$_2$ adsorption is exothermic on Pd (111), Pd$_3$S(202), and Pd$_4$S(200) surfaces (−2.51, −0.48, and −0.24 eV, respectively). Note that on Pd both the alkyne and H$_2$ compete for the same site, while on Pd$_3$S this is not the case because H atoms bind to surface sulfur atoms and the organic reactant at the Pd-ensembles. This bifunctional mechanism, where fragments are stored at different sites of the catalyst

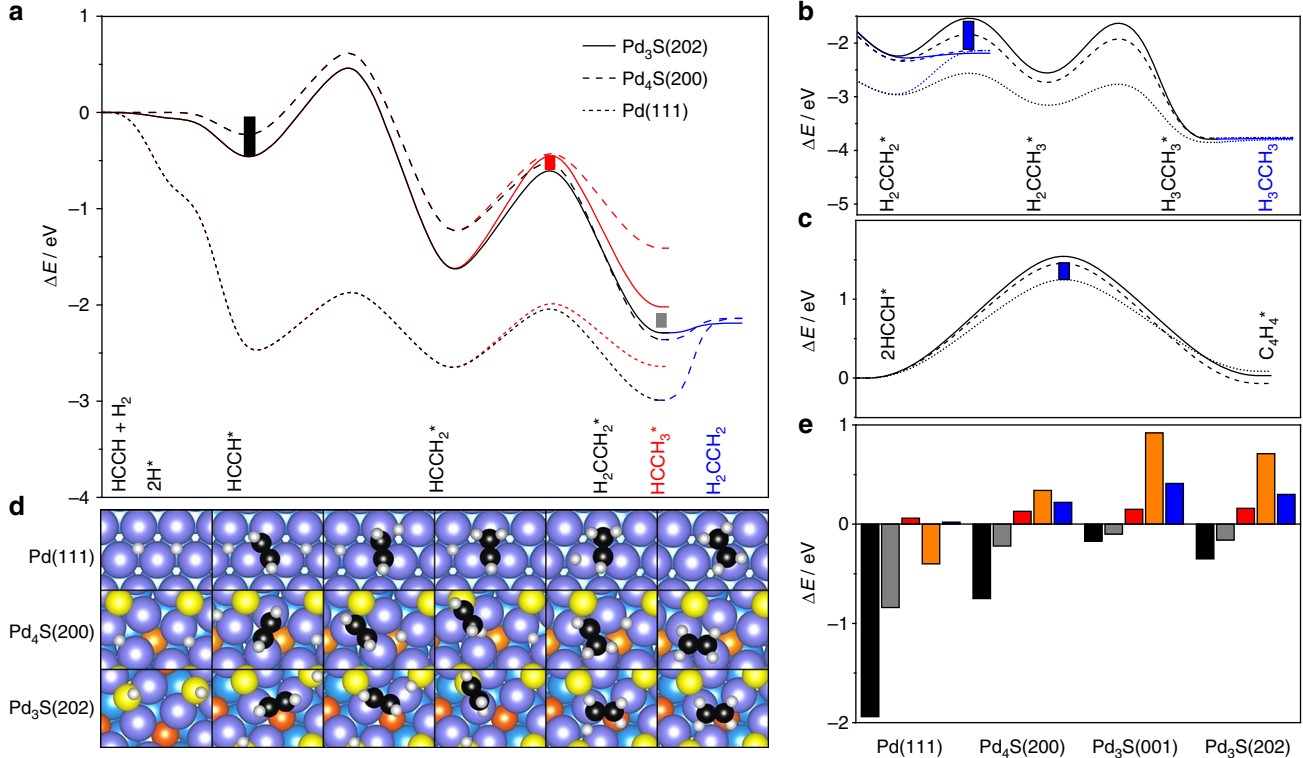

**Fig. 5** Density functional theory reaction analysis. **a–c** Energy profiles for the semi-hydrogenation (**a**), over-hydrogenation (**b**), and oligomerization (**c**) of acetylene on $Pd_3S(202)$, $Pd_4S(200)$, and $Pd(111)$. The asterisks denote adsorbed species, the color of the profile matches that of the possible intermediate, and the bars relate to the energy differences for $Pd_3S(202)$ presented in **e**: HCCH adsorption (black bars), $H_2CCH_2$ desorption (gray bars), and the difference in activation energy between $H_2CCH_2$ (ethene) and $HCCH_3$ (ethylidene) formation (red bars), $H_2CCH_2$ desorption and $H_2CCH_3$ formation (orange bars), and for oligomerization (blue bars). **d** Top view of the DFT-optimized adsorption configuration of the reaction intermediates and transition states (TS) on the surfaces. The elementary steps are shown in Supplementary Table 5. Color code as in Fig. 3, carbon (black), hydrogen (white)

surface, ensures higher activity of $Pd_xS$ than the reference $Pd/Al_2O_3$ (Supplementary Note 2). The first addition of H to the activated alkyne leads to a vinyl moiety ($HCCH_2$), whereas the second can result in either ethene ($H_2CCH_2$) or ethylidene ($HCCH_3$) formation. Notably, the difference in energy barrier between $H_2CCH_2$ and $HCCH_3$ formation on $Pd_3S(202)$ is greater than on $Pd(111)$ and on $Pd_4S(200)$. At this stage, the desired semi-hydrogenation product ($H_2CCH_2$) could desorb or undergo over-hydrogenation via an ethyl intermediate ($H_2CCH_3$) to form ethane. Ethene desorption from the $Pd_3S(202)$ and $Pd_4S(200)$ surfaces (0.08 and 0.22 eV, respectively) is energetically preferred over its further hydrogenation (0.79 and 0.55 eV, respectively), even when the zero point vibrational energy corrections are included (Supplementary Table 6), which is completely opposite to the behavior encountered on $Pd(111)$ where the over-hydrogenation barrier 0.45 eV is much lower than that of the alkene desorption (0.85 eV). This ultimately explains the low selectivity to ethene on Pd. The second potential side reaction investigated is oligomer formation, which was studied by modeling ethyne-ethyne coupling (Fig. 5c) and alkyne-vinyl paths providing similar insights due to scaling relations[62] (Supplementary Fig. 19, 21). The barrier for oligomerization is 1.34 eV on the $Pd(111)$ surface, and 1.64 eV on $Pd_3S(202)$, which indicates that the surface anisotropy hinders the diffusion of the alkyne on the latter surface, consistent with the expected site isolation. This explains why almost no oligomer formation is observed over the sulfided catalyst. Complementary analysis of the reaction profiles on the $Pd_3S(001)$ and $Pd_4S(110)$ surfaces are provided in Supplementary Note 4.

Long-term tests conducted to verify the stability of the $Pd_3S/C_3N_4$-423 and $Pd_4S/CNF$ catalysts, evidenced no signs of deactivation after 50 and 25 h on stream, respectively (Fig. 6a). Unfortunately, a longer test on the $Pd_4S/CNF$ sample was not possible due to the development of substantial pressure drop in the reactor, which resulted from the mechanical instability of the CNF. Analysis post-reaction by HRTEM, STEM, and XPS confirmed the similar composition and absence of nanoparticle sintering or phase segregation with respect to the fresh catalyst for $Pd_3S/C_3N_4$-423 (Fig. 2, Supplementary Fig. 5). This is an important advantage with respect to intermetallic compounds, where segregation effects have been reported to limit the ensemble control on the surface[20,44,46,63,64]. To shed light on the intrinsic stability of the system, detailed simulations were conducted to assess the likelihood of vacancy formation, sulfur incorporation to the reacting alkyne, and segregation effects. The energy penalty for forming a vacancy and releasing $H_2S$ on the (202) and (001) surfaces was calculated to be 0.56 and 1.58 eV, respectively, pointing to the stability under liquid-phase processes, while the formation energy of $H_2C = CHSH$ on the (202) and (001) surfaces is 1.13 and 2.60 eV, highlighting the impossibility of transferring sulfur to the substrate. Also, the segregation of a Pd atom from the bulk towards the surface requires 2.90, 4.30, and 2.96 eV for $Pd_3S(001)$, $Pd_3S(202)$, and $Pd_4S(200)$, respectively, meaning that even in the presence of high amounts of atomic H on the surface no preferential Pd or S segregation occurs[44]. Similarly, the formation of islands is not favored as the energy to exchange a Pd by a surface S atom is 3.12, 2.20, and 1.43 eV, respectively.

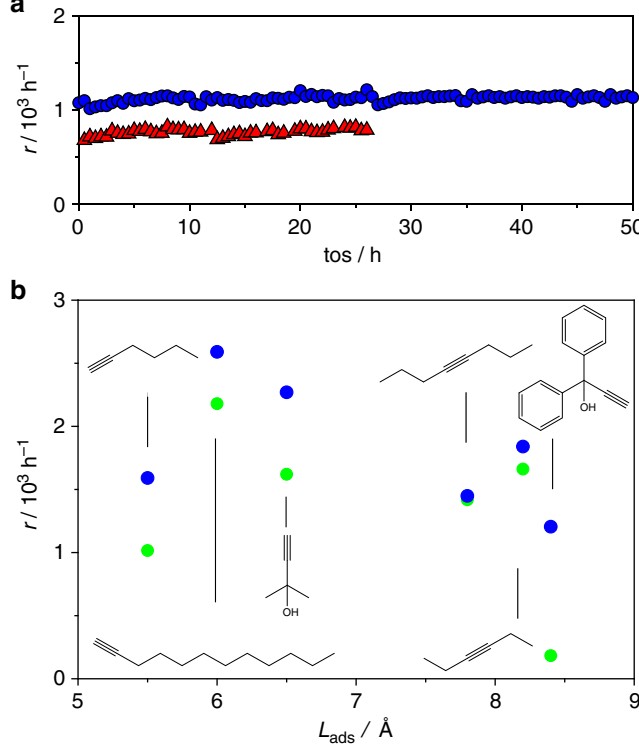

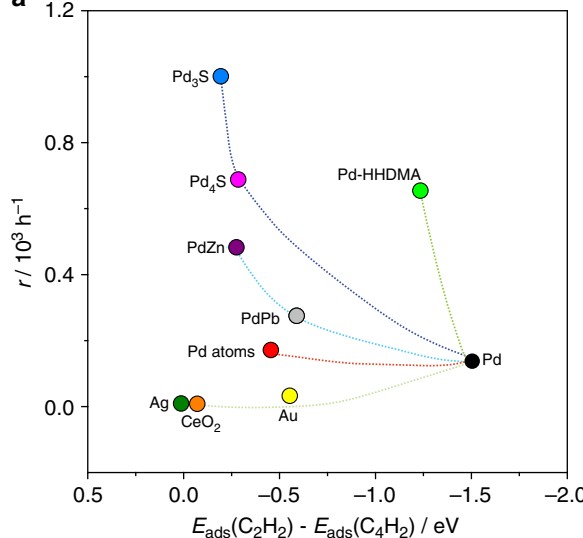

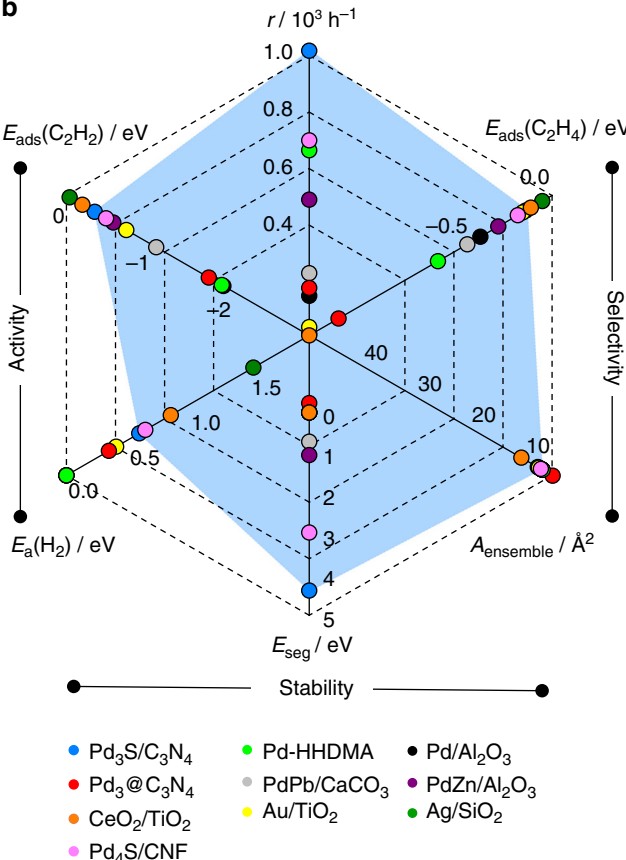

**Fig. 6** Stability and substrate scope of the palladium sulfide catalysts. **a** Both Pd₃S/C₃N₄ (blue circles) and Pd₄S/CNF (red triangles) display a stable performance on stream in the selective hydrogenation of 2-methyl-3-butyn-2-ol. **b** The impact of the adsorption length ($L_{ads}$) of functionalized alkynes of different sizes over Pd₃S/C₃N₄-423 (blue circles) and Pd-HHDMA (green circles). Conditions: $W_{cat} = 0.1$ g, $T = 303$ K, $P = 1$ (**a**) or 3 (**b**) bar, $F_L$(alkyne + toluene) = 1 cm³ min⁻¹, $F_G$(H₂) = 36 cm³ min⁻¹

Most of the relevant alkynes for the pharmaceutical and fine chemical industries are highly branched compounds with internal triple bonds and additional functionalities. To explore the scope of supported Pd₃S systems, the performance of Pd₃S/C₃N₄-423 has been compared with that of the commercial ligand-modified Pd-HHDMA catalyst in the hydrogenation of alkynes of varying size and functionality, namely, 1-hexyne, 3-hexyne, 1-dodecyne, and 1,1-diphenyl-2-propyn-1-ol (Fig. 6b). With adsorption lengths (defined by the size of the activated alkyne) shorter than 8 Å, similar rates were observed over both catalysts. However, while Pd-HHDMA becomes significantly less active at longer adsorption lengths, the performance of Pd₃S/C₃N₄-423 is independent of the size and functionality of the substrate. This effect can be rationalized by the fact that the architecture of active ensembles in Pd₃S is two dimensional as in the Lindlar system, whereas the HHDMA-modified catalyst shows a three dimensional structure featuring hydrophobic pores that provide access to the alkynes only if they are terminal and not too bulky[47]. Thus, we can conclude that the Pd₃S/C₃N₄-423 catalyst matches and even exceeds the performance of Pd-HHDMA, illustrating the effectiveness of the sulfur modification for preparing superior catalysts with clear practical scope.

**Fig. 7** Catalytic descriptors for alkyne semi-hydrogenation. **a** Rate of alkene formation at $T = 303$ K and $P = 1$ bar versus the differential adsorption energy of acetylene and ethene for all classes of catalysts. **b** Radar plot charting the relation of the rate with key theoretical indicators for the activity: the adsorption energy of the alkyne ($E_{ads}$(C₂H₂)) and the activation energy for H₂ dissociation ($E_a$(H₂)), selectivity: the alkene adsorption energy ($E_{ads}$(C₂H₄)), and the ensemble area ($A_{ensemble}$), and the stability: the segregation energy ($E_{seg}$). Values closest to the perimeter lead to enhanced properties. The blue shaded area connects the optimal values to date found for the Pd₃S system

## Discussion

To overcome the intrinsic limitations of palladium nanoparticles (i.e., uncontrolled ensemble size and strong adsorption of the alkene product), researchers have mainly targeted the integration of confined ensembles (no more than 3 atoms) and tailored

electronic properties (Fig. 1). To compare those strategies with $Pd_xS$ catalysts, we first evaluated the alkyne semi-hydrogenation performance using the differential adsorption energies, $E_{ads}(C_2H_2) - E_{ads}(C_2H_4)$, as a unified descriptor accounting for the intrinsic thermodynamic selectivity (Fig. 7a). Notice that other previously reported criteria for gas-phase hydrogenation screening such as the energy of methyl group adsorption[38] or the difference between the hydrogen barrier and the adsorption energy of the alkene[65] do not provide a satisfactory description. The former parameter does not consider the effects of van der Waals contributions, while the latter parameter is not suitable for catalysts in which hydrogen dissociation is rate determining (e.g., oxides). Even though all the modified materials display favorable values with respect to bare palladium (becoming more positive), correlation with the observed rates of 2-methyl-3-buten-2-ol formation reveals a wide variation in the productivity depending on the catalyst family. Thus, a framework and hierarchy of property-performance relationships were identified to account not only for selectivity, but also activity and stability (Fig. 7b).

Selectivity depends on (i) the adsorption energy of the alkene ($E_{ads}(C_2H_4)$) which shows an optimal value close to 0 or positive, and (ii) the ensemble area whose improved value (ca. 8 Å$^2$) is shared by all catalysts. Activity is driven by two criterion with different relative weight: the adsorption of the alkyne (described by the negative $E_{ads}(C_2H_2)$) is favorable over all catalysts, while most importantly the hydrogen activation energy ($E_a(H_2)$) discriminates between materials able of readily split molecular hydrogen and catalysts requiring harsh reaction conditions. Consideration of these factors clearly emphasizes the superiority of Pd-based systems, exhibiting the lowest barriers for $H_2$ dissociation and strong alkyne adsorption energies. Finally, stability could be assessed by several potential phenomena and scenarios, but for the sake of simplicity the segregation energy ($E_{seg}$), which is the differential energy between the ground state (i.e., an alloyed metallic surface) and the system after segregating one atom towards the surface, is taken as a measure for the robustness of the catalyst.

Compared to the archetypal Lindlar catalyst, sulfur in $Pd_xS$ systems would embed both the functions of acting as a molecular modulator on the ensemble mimicking the cooperative lead-quinoline role, and the electronic poisoning effect of lead[42]. However, sulfur poisoning is much more effective as very small nanoparticles with a high surface/volume ratio can be used instead of the large nanoparticles required for the PdPb catalyst. The need for facile hydrogen activation limits the practical relevance of oxides and other metals (Ag, Au)[19,26,34,35,66]. This is not the case in $Pd_xS$, intermetallics, ligand-modified nanoparticles, and single-atom catalysts due to the intrinsic ability of Pd to activate $H_2$. The excellent site isolation in well-defined ensembles featured by intermetallics, ligand-modified nanoparticles, and single-atom catalysts lead to quite selective catalysts[16,23,25,31]. However, the preferential hydrogen adsorption on Pd in alloys and intermetallics induces severe segregation effects spoiling the control of surface ensembles achieved via the precise synthesis[45,46]. Note that induced segregation issues are much more relevant in gas-phase applications due to the high temperature, pressure, and absence of an organic solvent (Supplementary Table 8). Remarkably, Pd segregation or islanding by substituting a palladium atom with sulfur is not favorable due to the polarization of the Pd-S bond enhancing the robustness of $Pd_3S$. This is even more enhanced in $Pd_3S(001)$ and $Pd_4S(110)$ as both display vacancy formation energy higher than $Pd_3S(202)$, making $Pd_4S$ more suited to high-pressure gas-phase operation. That is the reason why among all catalysts the $E_{seg}$ (accounting for the energy needed to enlarge the surface ensemble by one Pd atom from the bulk) of $Pd_3S$ and $Pd_4S$ are the highest ones (5.0

and 3.0 eV, respectively), while in PdPb this requires 0.7 eV. When considering ligand-modified systems like Pd-HHDMA, the three dimensional structure of such catalysts imposes accessibility constraints limiting their applicability for the conversion of bulky and internal alkynes, important intermediates in the fine chemical industry. This limitation is not present in the $Pd_xS$ catalyst as a two dimensional structure ensures the conversion of complex sterically hindered substrates. Single atoms display distinct geometric and electronic properties compared to catalysts based on nanoparticles, while the scaffold can also contribute to $H_2$ activation and stabilization of the reactants[31]. Sulfur atoms in $Pd_xS$ also share some of these features. The adsorbed H atoms are stored at the S centers in the form of thiols leaving the Pd site free for alkyne adsorption. This means that there is no competition between the alkyne and hydrogen for site availability and the mechanism is similar to that of a dual site, which is again a consequence of the Pd-S bond polarity. The bifunctionality is more effective as no competition between the alkyne and hydrogen for the active sites occurs, this is reflected in the different reaction orders with respect to the alkyne, which is negative for Pd and close to zero for $Pd_3S$ (Supplementary Note 2). This explains the higher activity of $Pd_3S/C_3N_4$ than the reference Pd/$Al_2O_3$. Therefore, the combination of crystal phase and orientation control observed in the sulfidation process to form the $Pd_3S$/$C_3N_4$ catalyst ensures that the best values for the set of descriptors are achieved in an elegant, yet efficient manner constituting a revolutionary step in catalyst design.

In conclusion, supported palladium sulfides have been shown to possess unique ensembles with face- and phase-dependent geometric and electronic characteristics and high stability, demonstrating exciting potential as catalysts for the selective hydrogenation of alkynes. A mild sulfidation treatment over supported metallic Pd nanoparticles led to the incorporation of sulfur into the palladium lattice, which drives the formation of a nanostructured $Pd_3S$ phase preferentially exposing the (202) crystal facets. This surface displays isolated $Pd_3$ ensembles surrounded by sulfur atoms featuring the right ensemble and electronic poisoning to allow the selective hydrogenation of linear, branched, and bulky functionalized alkynes. Ensembles of similar architecture and functionality were also evidenced in $Pd_4S/CNF$. The $Pd_xS$ catalysts outperformed all state-of-the-art catalysts in terms of alkene formation rate per mole of Pd, and evidenced high durability with no sign of segregation. A detailed comparison to the existing hydrogenation catalysts illustrated the hierarchy of key performance descriptors, demonstrating that the reported synthetic protocol produces a catalyst that synergistically combines the best characteristics of bond polarization, crystal phase and orientation control, while keeping a two-dimensional nature enabling the superior performance for a broad set of compounds. The present approach illustrates how the concept of ensemble control in kinetically-trapped crystal orientations can be employed in the precise design of evolved catalysts.

## Methods

**Catalyst preparation**. The mesoporous graphitic carbon nitride carrier ($C_3N_4$) was prepared by controlled polymerization. Briefly, cyanamide (3.0 g, Sigma-Aldrich, 99%) and an aqueous dispersion of colloidal silica (12 g, 40 wt.% solids, 12 nm average diameter, Sigma-Aldrich) were mixed and stirred at 373 K for 6 h to completely evaporate the water. The resultant solids were ground and calcined at 823 K (2.3 K min$^{-1}$ ramp) for 4 h under a nitrogen flow (15 cm$^3$ min$^{-1}$). The silica was removed by treating the solid in a 4 M aqueous solution of $NH_4HF_2$ (200 cm$^3$, Acros Organics, 95%) for 48 h. The template free mesoporous carbon nitride was collected by filtration, washed thoroughly with distilled water and ethanol, and dried at 333 K overnight.

The incorporation of palladium nanoparticles was approached by wet deposition, targeting a metal loading of 0.5 wt.%. The carrier was dispersed in deionized water (0.01 g$_{carrier}$ cm$^{-3}$) under sonication for 1 h prior to the introduction of an aqueous solution (4.72 cm$^3$ g$_{carrier}$$^{-1}$) containing $PdCl_2$ (0.01 M,

ABCR Chemicals, 99%) and NaCl (0.04 M, Sigma-Aldrich, 99%). The resulting suspension was stirred at room temperature for 2 h in the dark. To promote metal nanoparticle formation, an aqueous solution (1.4 cm$^3$ g$_{carrier}$$^{-1}$) of NaBH$_4$ (0.5 M, Sigma-Aldrich, 99%) was subsequently added dropwise and the mixture was stirred overnight. Finally, the solids (Pd/C$_3$N$_4$) were collected by filtration, washed thoroughly with distilled water and ethanol and subsequently dried at 333 K overnight. The atomic dispersion of palladium on C$_3$N$_4$ (Pd@C$_3$N$_4$) was achieved by microwave-assisted-deposition. An aqueous solution of Pd(NH$_3$)$_4$(NO$_3$)$_2$ (0.05 cm$^3$, 5 wt.% Pd, ABCR Chemicals) was added to a suspension (20 cm$^3$) of the dispersed C$_3$N$_4$ (0.5 g) and stirred overnight in the dark. The resulting suspension was treated in a microwave reactor (CEM Discover SP), applying a cyclic program of 15 s irradiation and 3 min cooling with 20 repetitions using a power of 100 W. The resulting powder was collected, washed, and dried as described for the nanoparticle-based system.

The catalyst was added to a Teflon-lined autoclave containing an aqueous solution of Na$_2$S·9H$_2$O (Sigma-Aldrich, 99%) using a molar S/Pd ratio of 10. The autoclave was then heated to the desired temperatures (373, 398, or 423 K) for 3 h. The solid was collected by filtration, extensively washed with deionized water and ethanol, and dried overnight at 333 K.

In an alternative route, a supported Pd$_4$S catalyst was prepared by incipient wetness impregnation of PdSO$_4$ (ABCR-Chemicals, 99.99%) on CNFs (Aldrich). After drying the resulting material was treated under 10 vol.% H$_2$/He at 523 K for 1 h.

Reference PdPb/CaCO$_3$ (Alfa Aesar, ref: 043172) and Pd-HHDMA, where HHDMA stays for hexadecyl(2-hydroxyethyl)dimethylammonium dihydrogen phosphate (NanoSelect LF 200$^{TM}$, Strem Chemicals, ref.[46]-1711) catalysts were used as received. No quinoline was added to the PdPb/CaCO$_3$ catalyst during testing.

**Characterization methods.** The Pd and Na contents in the catalyst were determined by inductively coupled plasma-optical emission spectrometry (ICP-OES) using a Horiba Ultra 2 instrument equipped with a photomultiplier tube detector. The samples were dissolved in piranha solution under sonication until the absence of visible solids. The S content was determined by infrared spectroscopy using a LECO CHN-900 combustion furnace. Nitrogen isotherms were obtained at 77 K using a Micrometrics Tristar instrument, after evacuation of the samples at 423 K for 3 h. The Brunauer–Emmett–Teller (BET) method was applied to calculate the total surface area ($S_{BET}$). High-resolution transmission electron microscopy (HRTEM), high-angle annular dark field STEM (HAADF-STEM), and elemental mapping with EDX spectroscopy were conducted on a FEI Talos microscope operated at 200 kV. Samples were prepared by dusting respective powders onto lacey-carbon coated nickel grids. The particle size distribution was assessed by counting more than 100 individual Pd nanoparticles. CO chemisorption was performed using a Micrometrics Autochem II 2920 chemisorption analyzed coupled with a MKS Cirrus 2 quadrupole mass spectrometer. The sample was pretreated in situ at 393 K under He flow (20 cm$^3$ min$^{-1}$) for 60 min, and reduced at 348 K under flowing 5 vol.% H$_2$/N$_2$ (20 cm$^3$ min$^{-1}$) for 30 min. Thus, 0.491 cm$^3$ of 1 vol.% CO/He was pulsed over the catalyst bed at 308 K every 4 min. The palladium dispersion was calculated from the amount of chemisorbed CO, considering an atomic surface density of $1.26 \times 10^{19}$ atoms m$^{-2}$ and an adsorption stoichiometry Pd/CO = 2. Powder XRD was measured using a PANalytical X'Pert PRO-MPD diffractometer and Cu-K$\alpha$ radiation ($\lambda = 0.154$ nm). The data was recorded in the 10–70° 2$\theta$ range with an angular step size of 0.017 and a counting time of 0.26 s per step. XPS was conducted with a Physical Electronics Instruments Quantum 2000 spectrometer using monochromatic Al K$\alpha$ radiation generated from an electron beam operated at 15 kV and 32.3 W. The spectra were recorded under ultra-high vacuum conditions (residual pressure = $5 \times 10^{-8}$ Pa) at a pass energy of 50 eV. In order to compensate for charging effects, all binding energies were referenced to the C 1$s$ at 288.2 eV. $^{13}$C solid-state cross-polarization/magic angle spinning nuclear magnetic resonance (CP/MAS NMR) spectra were recorded on a Bruker AVANCE III HD NMR spectrometer at a magnetic field of 16.4 T corresponding to a $^1$H Larmor frequency of 700.13 MHz. A 4 mm double resonance probe head at a spinning speed of 10 kHz was used for all experiments. The $^{13}$C spectra were acquired using a cross polarization experiment with a contact time of 2 ms and a recycle delay of 1 s. A total of $64 \times 10^3$ scans were added for each sample. $^{13}$C experiments used high-power $^1$H decoupling during acquisition using a SPINAL-64 sequence.

**Catalytic testing.** The hydrogenation of 2-methyl-3-butyn-2-ol (Acros Organics, 99.9%), 1-hexyne (Acros Organics, 98%), 3-hexyne (Acros Organics, 99%), 1-dodecyne (Acros Organics, 98%), 4-octyne (Acros Organics, 98%), and 1,1-diphenyl-2-propyn-1-ol (Sigma-Aldrich, 99 %) was carried out in a fully-automated flooded-bed reactor (ThalesNano H-Cube Pro$^{TM}$), in which the gaseous hydrogen, produced in situ by the electrolysis of Millipore water, and the liquid feed flow concurrently upward through a cartridge of approximately 3.5 mm internal diameter. The latter contains a fixed bed which is composed of the catalyst (0.10 g) well mixed with silicon carbide as diluent (0.12 g, Aldrich, 99.8%), both with particle size of 0.2–0.4 mm. The potential role of heat- and mass-transfer limitations was excluded using the Koros-Nowak diagnostic test (Supplementary Note 2). The liquid feed contained 5 wt.% of substrate, 3 wt.% of benzene (Merck,

99%) as internal standard, and toluene (Fischer Chemicals, 99.95%). The reactions were conducted at various temperatures (303–363 K) and total pressures (1–8 bar), keeping constant liquid (1 cm$^3$ min$^{-1}$) and H$_2$ (36 cm$^3$ min$^{-1}$) flow rates. The reaction orders were obtained as a function of the gas-phase H$_2$ pressure and the 2-methyl-3-butyn-2-ol concentration. Blank alkyne semi-hydrogenation experiments over the range of conditions investigated, excluded the competitive hydrogenation of benzene or toluene. The stability test was carried out at $T = 303$ K and $P = 1$ bar. The products were collected after 20 min of steady-state operation. The batch hydrogenation was carried out in a CEM Discover SP microwave reactor with a pressure-controlled vessel under continuous stirring. In a typical reaction, the liquid feed (1.5 cm$^3$) was heated in the presence of the catalyst (0.01–0.02 g) at 323 K. The initial H$_2$ pressure was 3 bar. In all cases, the products were analyzed offline using a gas chromatograph (HP-6890) equipped with a HP-5 capillary column and a flame ionization detector. The conversion ($X$) of the substrate was determined as the amount of reacted substrate divided by the initial amount of substrate. The reaction rate ($r$) was expressed as the number of moles of 2-methyl-3-buten-2-ol produced per hour and total moles of Pd. The selectivity ($S$) to each product was quantified as the amount of the particular product divided by the amount of reacted substrate. The carbon balance exceeded 98% in all cases, excluding the formation of oligomers. Consistently, negligible differences were observed between the thermogravimetric profiles of both the fresh and used catalyst.

**Computational details.** The reaction network for hydrogenation on Pd and Pd$_x$S surfaces was investigated by means of DFT calculations with periodic boundary conditions[67] within the RPBE approach with a plane wave cut-off energy of 450 eV (for more details see Supplementary Method 1 and Note 4) and dispersion contributions with D2 parameters[68,69] complemented with our reparametrized values for metals[70]. The models for the Pd$_3$S and Pd$_4$S surfaces were constructed with six S-Pd-S trilayers where the two topmost S-Pd-S trilayers were fully relaxed, whereas the four bottommost layers were fixed to their bulk positions. The bare Pd(111) surface was modeled as a periodically repeated $p(3 \times 2)$ slab consisting of five atomic layers, where the three at the bottom were fixed and the two on the top were fully relaxed. Due to the operation in liquid phase and the low binding energies found for the reactants, no significant deviation from the original stoichiometry is observed, which differs from gas-phase operation[59]. The climbing image-modified nudged elastic band (CI-NEB) method[71,72] was employed to locate the transition states. To date, the solvent contributions to the evaluation of Gibbs energies is still under debate, thus, the energy profiles are presented in terms of the potential energies. The contributions of zero-point vibrational energy (Supplementary Table 6) and vibrational entropies to competing steps are similar for bifurcation steps, thus their differences are small enough to be safely neglected. The computed structures can be retrieved from the ioChem BD database[73] via the link (https://doi.org/10.19061/iochem-bd-1-57).

**Data availability.** The authors declare that the data supporting the findings of this study are available within the article and its Supplementary Information file. All other relevant source data are available from the corresponding author upon reasonable request.

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

## Acknowledgements

Financial support from ETH Zurich, the Swiss National Science Foundation (Grant No. 200021-169679), and MINECO (grant number CTQ2015-68770-R). The Barcelona Supercomputing Centre (BSC-RES) and ScopeM of ETH Zurich are thanked for access to their facilities. Dr René Verel is thanked for help with the MAS NMR measurements. M. S. thanks the COFUND program and MINECO for support through Severo Ochoa Excellence Accreditation 2014-2018 (SEV-2013-0319).

## Author contributions

J.P.-R. and N.L. conceived and coordinated all stages of this research. D.A. and Z.C. prepared and characterized the samples. D.A. performed the catalytic tests. S.M. acquired the microscopy images. R.H. undertook the XPS analyses. M.S. conducted the DFT simulations. All authors contributed to writing the manuscript.

## Additional information

**Competing interests:** The authors declare no competing interests.

