## [Peer Review File · Nature Communications]

Reviewers' comments:

Reviewer #1 (Remarks to the Author):

The authors have presented an approach to design selective semi-hydrogenation catalyst by using S as a modifier to not merely decorate the surface (which is the traditional approach for partial poisoning via CO or H₂S) but rather modify the crystal structure of the catalyst. The work is dominated (and led) by DFT calculations, and the experiments amount to measurements of activity and selectivity over three Pd-based catalysts. There are numerous gaps in the manuscript; experimental and computational details are missing. The development of a descriptor and its description in the text was ineffective. The ultimate descriptor doesn't really work that well for the small group of catalysts tested. The strength of the paper lies in the unique reactivity of the sulfide Pd catalyst, but a complete understanding of why is lacking and diluted out by the excessive content on the "other" Pd catalysts.

There is no doubt this manuscript needs work before it can be considered for publication in Nature Communications. This manuscript must be reviewed a second time before the manuscript should be considered for acceptance in Nature Communications.

1. The word "ensemble control" and "design" are ABSOLUTE overstatements and should be removed from the title. These words give a false perception of what is actually demonstrated in the manuscript. Our understanding is that the authors do not have any real control on the active site nuclearity and rather than employing a design strategy, they are at the mercy of the way the material crystallizes. This is NOT design since you have no control. This is merely thermodynamics and limited exploration of the current Pd-S system! Further, it does not appear this method provides any availability to access nuclearities other than three.
2. It is unclear how a selective poison ensures facile activation of reactants. One could rationalize selective (one functional group over another) activation, but facile seems counter-intuitive.
3. The DFT calculations suggest the adsorption energy of ethylene on PdS(202) is only 0.08 eV (8 kJ/mol). This is an exceedingly low adsorption energy. It seems too low! Does ethylene only physisorb to the surface via vdW attractions? Is a di-sigma or pi-bonded species preferred?
4. Figure 6a and b needs much more detail than has been provided. Where does this data come from? Are all the adsorption energy difference between acetylene and ethylene calculated by the

authors in Figure 6a? How was the difference in adsorption energy (at 0 K) calculated for Pd-HHDMA calculated? How was the polymer HHDMA represented in the DFT calculation? Similar questions of where the data actually came from is apparent for Figure 6b too.

5. In the comparison of the catalytic activity, did the authors take any particular measure to assess the role of mass transfer limitations? There is no discussion of an evaluation of potential mass transfer effects (Madon-Boudart, Koros-Nowack, etc.).

6. The choice of solvent (benzene and/or toluene) is somewhat puzzling from the standpoint that both of these solvents can be hydrogenated. The experimental temperature are probably too low, but have the authors confirmed there is no competitive hydrogenation? It would be interesting to determine if an ensemble size of Pd₃ is sufficient to hydrogenate arenes.

7. It is unclear from the description in the experimental section if the turnover frequency (TOF) was calculated based on the total moles of Pd or the surface moles of Pd exposed (i.e., dispersion) in each case.

8. Equation 1 of SI to calculate surface energy doesn't look correct. Why is the surface energy of an unrelaxed surface subtracted from the bulk energy? Can the authors please explain the reason behind this approach for the calculation of surface energy?

9. Regarding the calculation of surface energy, there will be some termination that non-stoichiometric. In the case of non-stoichiometric terminations, the subtraction of the bulk energy seems like a gross averaging of energies of Pd and S and perhaps an over simplification.

10. The (202) termination of the material is expected to be a significant fraction of the particle surface according to DFT calculations. This facet also has hydrogen "storage" capacity through thiol formation. There has been extensive reporting that "stored" hydrogen in the form of beta-hydrides are more active than ambient hydrogen dissociated on the surface. We recognize thiol H is chemically distinct but has any mechanistic calculations been conducted to identify the propensity of this hydrogen to affect the reaction energy diagram?

11. An appropriate and worthwhile addition to this manuscript is a Wulff construction of a Pd₃S particle using the surface energies listed in Table S2.

12. A popular mechanism for oligomerization is the reaction of a partially dissociated acetylene (i.e., vinyl) with acetylene rather than the coupling of two acetylene molecules. What is the motivation for choosing a dimerization reaction pathway between acetylene? The reference by Spanjers et al.¹ utilizes a surface reaction between adsorbed acetylene and vinyl to form C₄ products.

13. Regarding the evidence of surface segregation for intermetallic compounds we would recommend the authors cite the HS-LEIS work by Rameshan et al.²⁻⁴
14. The authors comment on the activity of different catalysts for different alkyne semi-hydrogenations. The presentation of data is incomplete and the authors should comment on the selectivity and product distribution (cis, trans E, Z etc).
15. The authors make a mechanistic claim regarding the dual site nature of the reaction pathway with regards to alkyne and hydrogen binding. The authors do NOT provide enough experimental evidence to make this statement with absolute certainty. This is a very long-standing debate in the community (at least for acetylene, which the authors use as the model compound for making these assertions through DFT) and surely further experiments and microkinetic modeling must be conducted before the mechanism can be fully established.
16. The author's claim regarding the resistant nature of these materials to surface segregation is tenuous at best. Though the energetics from DFT support a lack of surface segregation, these calculations are conducted in vacuum. It is important to note that surface segregation is a very strong function of the interface chemical composition (presence of O₂, H₂ etc) and temperature. These conditions are typically not captured through DFT surface calculations. Refer to HS-LEIS work by Stadlmayr et al.⁴ and Kopfle et al.⁵ as examples.
17. The section of the manuscript "A long-term test conducted...ensemble control of the surface" does not present a coherent story. We doubt there is any direct one to one correlation between phase separation (presumably the authors are only probing the bulk via XRD?), sintering and compositional changes (once again, the bulk via ICP or STEM?) with the surface composition and ensemble size control. The fact that the material has a stable activity does not imply there is no surface segregation as this phenomenon can occur quite rapidly (at the instant the material is exposed and before the first point is collected).
18. Was there any attempt (such as TGA-MS) to quantify C deposition. What does a carbon mass balance in this reaction look like?
19. We would recommend the authors to cite the relevant paper from Yang et al.⁶
20. Figure 1 is odd and irrelevant. It is more suited for a review paper, which this is not. It is irrelevant because it does not add to the understanding of the molecular origins of the observed activity-selectivity in this system.
21. The first sentence of the paper makes no sense. A 'heterogeneous catalyst' not

'heterogeneous catalysis' feature a high density of isolated sites.... The first sentence must be rewritten!...

22. What the authors referring to when they believe a PdS surface mimics the "enzymatic strategy of electronic density control"? I am well versed in mechanistic enzymatic catalysis and this statement has little concrete meaning to me. The authors should provide a reference and at a minimum, a description of how PdS approximates electron distribution (?) observed in enzymes.

23. What is the motivation of choosing 2-methy-3-butyn-2-ol as a model substrate?

References

1. Spanjers, C. S.; Held, J. T.; Jones, M. J.; Stanley, D. D.; Sim, R. S.; Janik, M. J.; Rioux, R. M., Zinc inclusion to heterogeneous nickel catalysts reduces oligomerization during the semi-hydrogenation of acetylene. *Journal of Catalysis* 2014, 316, 164-173.
2. Rameshan, C.; Stadlmayr, W.; Penner, S.; Lorenz, H.; Mayr, L.; Hävecker, M.; Blume, R.; Rocha, T.; Teschner, D.; Knop-Gericke, A.; Schlögl, R.; Zemlyanov, D.; Memmel, N.; Klötzer, B., In situ XPS study of methanol reforming on PdGa near-surface intermetallic phases. *Journal of Catalysis* 2012, 290 (Supplement C), 126-137.
3. Rameshan, C.; Stadlmayr, W.; Weilach, C.; Penner, S.; Lorenz, H.; Hävecker, M.; Blume, R.; Rocha, T.; Teschner, D.; Knop-Gericke, A.; Schlögl, R.; Memmel, N.; Zemlyanov, D.; Rupprechter, G.; Klötzer, B., Subsurface-Controlled CO₂ Selectivity of PdZn Near-Surface Alloys in H₂ Generation by Methanol Steam Reforming. *Angewandte Chemie International Edition* 2010, 49 (18), 3224-3227.
4. Stadlmayr, W.; Rameshan, C.; Weilach, C.; Lorenz, H.; Hävecker, M.; Blume, R.; Rocha, T.; Teschner, D.; Knop-Gericke, A.; Zemlyanov, D.; Penner, S.; Schlögl, R.; Rupprechter, G.; Klötzer, B.; Memmel, N., Temperature-Induced Modifications of PdZn Layers on Pd(111). *The Journal of Physical Chemistry C* 2010, 114 (24), 10850-10856.
5. Kopfle, N.; Mayr, L.; Lackner, P.; Schmid, M.; Schidmair, D.; Gotsch, T.; Penner, S.; Klotzer, B., Zirconium-Palladium Interactions during Dry Reforming of Methane. *ECS Trans* 2017, 78 (1), 2419-2430.
6. Yang, B.; Burch, R.; Hardacre, C.; Hu, P.; Hughes, P., Selective Hydrogenation of Acetylene over Pd-Boron Catalysts: A Density Functional Theory Study. *Journal of Physical Chemistry C* 2014, 118 (7), 3664-3671.

Reviewer #2 (Remarks to the Author):

This paper contains information which will be of value to those working in the rational design of solid surfaces for selective catalysis. The work focusses on the use of two sulphur containing

structures which both lead to disruption to the Pd ensemble and modify adsorption characteristics of the adsorbates and intermediates. The data for both Pd₃S and Pd₄S is of value and the latter has already been shown to exhibit useful properties in selective hydrogenation as appropriately cited by the authors. However, the origin of the later work was based on an observation that S containing ligands, even after decomposition but with retention of a sulphur adatom/embedded atom, show useful properties in alkyne semi-hydrogenation. As the authors indicate (Fig 1) that they are interested in the historical evolution of these catalysts, this reference (or both) should be mentioned (Chem Commun., 2011, 47, 2351 - 2353 and J. Catalysis, 2011, 281, 231–240).

A few other points to address/provide comments:

The reaction work involved 3-methyl-3-butyn-2-ol which is a liquid phase reaction (usually in the presence of solvent) while the calculations are described for a more "simple" 2 carbon alkyne, acetylene (gas phase reaction). I understand why this has been done in terms of computational time/capacity however, the authors should justify why an extrapolation may be performed (we often find that catalysts which show promise for gas phase hydrogenation of eg acetylene are not so selective in liquid phase of hex/heptynes and visa versa).

Authors should make clear that Lindlar is usually employed in the presence of quinolone (which I am assuming, is not employed here).

p-8. There is an assumption (not substantiated) that excellent selectivity is a consequence of the absence of Beta hydride in the Pd₃S. Are the authors convinced that no hydride is formed (check refs 11 SI and 51 in main manuscript).

The hydrogen is deemed to be located on the S atoms following dissociation on Pd. Is there any spectroscopic evidence for these thiols? In the DFT work, I was unable to find an energy value given to the subsequent transfer of these hydrogen to the adsorbed intermediates. If hydrogenation or hydrogen transfer take place between a thiol and an adsorbed hydrocarbon, why cant we hydrogenate using a hydroxyl group on an oxide (normally...)?

What happens to the sodium which was used as precursor for addition of S? Is any retained? What form does it exist in, where is it located and what consequences does it have in terms of catalytic behaviour?

p-12. The authors discuss the issue of surface segregation in the context of stability however, given that the highest temperature employed is 363 K, do they believe the barrier to atom diffusion from the bulk is overcome under these conditions (irrespective of what the most thermodynamically stable surface may be)?

How does (SI, page 15) the calculated heat of adsorption of acetylene compare with the published experimental values?

The authors measure CO uptakes (Table 1 and SI table 1)) for samples including Pd₃S yet previous studies (ref 51) show that Pd₄S does not take up (appreciable) amounts of CO. Can the authors add a comment to account for this difference?

Reviewer #3 (Remarks to the Author):

The authors conducted studies on the semi-hydrogenation of alkynes over Pd₃S from both experimental and theoretical aspects. My comments are listed as follows.

1. As mentioned by the authors in the manuscript, Pd₄S type of Pd sulfide catalyst has been reported in the literature (e.g. ref. 31), but the direct comparison between the performance of the catalyst developed in this work and Pd₄S is missing. The activity, selectivity and stability of Pd₄S determined experimentally should be included in this work to support the computational results. In addition, it is strongly suggested that the authors state clearly new findings and main difference between this work and those from Anderson's group, otherwise the current work could only be considered as a follow-up work.

2. I cannot see why the authors tested the catalysts with liquid phase alkynes such as 2-methyl-3-butyn-2-ol, 1-hexyne and 3-hexyne, but did calculations with the simplest alkyne, i.e. acetylene, in gas phase. Although the authors claimed that acetylene "demonstrates a similar adsorption energy and configuration to 2-methyl-3-butyn-2-ol for all the Pd_xS catalysts considered", it is widely recognized that molecules in the liquid and gas phases are quite different from many aspects, such as the distances between molecules and the structure and entropy of molecules. Therefore, the adsorption free energies of acetylene are totally different from 2-methyl-3-butyn-2-ol under the reaction conditions.

3. The authors found that the "Pd₃S/C₃N₄ displays a stable performance for 50 h on stream in the selective hydrogenation of 2-methyl-3-butyn-2-ol". How is the stability of other catalysts the authors used as references, i.e. PdPb/CaCO₃, Pd-HHDMA, and Pd@C₃N₄?

4. This point is also related to the stability of the catalysts. The authors included STEM images in Fig.4c to show that Pd₃S displays a stable performance in the hydrogenation of 2-methyl-3-butyn-2-ol. However, they also attributed the high stability of Pd₃S/C₃N₄ catalyst to the higher coupling ethyne-ethyne barrier than Pd, indicating that oligomerization could be suppressed over Pd₃S, the point of which should also be justified experimentally in the manuscript.

5. According to the equation included in the SI that the authors used for the surface energy calculations, i.e. Eq. 1, it seems that the authors only considered the stability of different terminations of Pd₃S in vacuum. However, they should keep in mind that the catalyst is exposed to hydrogen and alkynes during reaction, and it is highly possible that the surface sulfur is hydrogenated to SH. This will definitely change the surface energies of different terminations obtained by the authors.

6. It seems that the adsorption of C₂H₂ over Pd₃S is too weak, even weaker than over Au. The transition state of C₂H₂ hydrogenation is even higher than the gaseous state in the energy profile shown in Fig. 5. In addition, while considering the adsorption free energies of H₂ and C₂H₂, one can anticipate that both adsorption are endogonic at the reaction temperature. Have the authors calculated the reaction rates theoretically based on the energies shown in Fig.5a to see whether these energies obtained from DFT calculations support the experimental results that Pd₃S is more active than Pd?

7. The usage of E_{ads}(C₂H₂)–E_{ads}(C₂H₄) as a descriptor should be justified.

8. The activation barrier difference between the formation of H₂CCH₂ and HCCH₃ are too small on all surfaces studied. Therefore, zero-point energy corrections and entropy effect should be considered.

NCOMMS-17-31323 - Response to Reviewers

Comments in *blue* - Replies in black - Actions in **bold**

Indicated figures and page numbers refer to the revised manuscript with changes highlighted

We warmly thank the Reviewers for their valuable feedback and have carefully considered the comments raised. This prompted us to generalize the concept of selective ensembles to a palladium sulfide with different stoichiometry, reinforcing the unique aspects of our approach to exploit the exceptional properties of *p*-block elements for the design of improved catalysts for the liquid-phase semi-hydrogenation of functionalized alkynes. Answers and actions taken upon revision are detailed below.

Reviewer #1

The authors have presented an approach to design selective semi-hydrogenation catalyst by using S as a modifier to not merely decorate the surface (which is the traditional approach for partial poisoning via CO or H₂S) but rather modify the crystal structure of the catalyst. The work is dominated (and led) by DFT calculations, and the experiments amount to measurements of activity and selectivity over three Pd-based catalysts.

The development of crystalline palladium sulfides instead of surface-decorated materials is certainly a distinguishing aspect of our work and theory underpins the molecular understanding gathered for these catalysts. However, we disagree that ‘the experiments only amount to measurements of activity and selectivity over three Pd-based catalysts’. Firstly, this neglects the introduction of a novel and scalable synthetic approach to prepare supported Pd₃S nanoparticles with a narrow size distribution and controlled orientation using graphitic carbon nitride as a carrier. Secondly, the catalytic evaluation was conducted over seven Pd-based catalysts mapping the influence of temperature and pressure in each case. Thirdly, in addition to evaluating the activity and selectivity of the palladium sulfide over a broad substrate scope, we also demonstrated the catalyst stability for 50 h on stream. **We have now taken this a step further by reproducing a recently reported synthetic protocol to prepare a Pd₄S supported on carbon nanofibers** (it was not possible to isolate this phase on carbon nitride). Detailed characterization of the resulting material identified similar trimeric ensembles in the dominant surface. Evaluation of the alkyne semi-hydrogenation performance also confirmed the exceptional selective properties of the Pd₄S phase. **We have also strengthened the catalytic tests**, evaluating the stability of all catalysts, establishing the comparative kinetic fingerprint of supported Pd₃S and Pd nanoparticles, and broadening the substrate comparison to include 1-hexyne for all the catalytic materials.

There are numerous gaps in the manuscript; experimental and computational details are missing.

We have carefully revised the manuscript to ensure that all relevant experimental and computational details are provided.

The development of a descriptor and its description in the text was ineffective. The ultimate descriptor doesn't really work that well for the small group of catalysts tested.

We sought to identify a universal descriptor for the performance of alkyne semi-hydrogenation catalysts. However, comparison of different property-performance relationships clearly revealed that a single governing criterion is elusive due to the breadth of the material demands. Consequently, we identify the key actors for each pillar of function (activity, selectivity, and stability), from which we are able to determine the hierarchy of the degree of optimization in the principal catalyst families (see **Figure 7**). **The Discussion has been rewritten to clarify this, identifying the relative strengths and weaknesses of each catalyst family (page 12, line 22)**. Based on this analysis, two parameters are identified to have the largest impact on the overall performance, the minimization of the barrier for the activation of molecular hydrogen and the stability of the catalyst ensemble.

The strength of the paper lies in the unique reactivity of the sulfide Pd catalyst, but a complete understanding of why is lacking and diluted out by the excessive content on the “other” Pd catalysts. There is no doubt this manuscript needs work before it can be considered for publication in Nature Communications. This manuscript must be reviewed a second time before the manuscript should be considered for acceptance in Nature Communications.

Thank you for highlighting the unique reactivity of our palladium sulfide catalysts. The new computational and experimental additions to the manuscript provide strong evidence that the trimeric ensembles in the dominant surface of the Pd₃S nanoparticles are responsible for the exceptional performance. In particular, they confirm that (i) hydrogen activation can readily occur *via* thiol formation, (ii) the presence of sulfur moderates the binding strength of both the alkyne and the alkene while preserving a positive thermodynamic selectivity, and (iii) the crystalline structure ensures a high stability of the surface configuration that prevents oligomerization. As mentioned above, the new results generalizing to a supported palladium sulfide phase with distinct stoichiometry, *viz.* Pd₄S, further corroborate the unique properties of the trimeric ensembles.

As the Reviewer will appreciate, to demonstrate this it is necessary to contextualize the findings, which justifies the comparison with other state-of-the-art materials. However, to avoid potential dilution of the understanding of the supported palladium sulfides, **we have now moved the microscopy images of the previously reported catalysts to Supplementary Figure 1 (page 3).**

1. The word “ensemble control” and “design” are ABSOLUTE overstatements and should be removed from the title. These words give a false perception of what is actually demonstrated in the manuscript. Our understanding is that the authors do not have any real control on the active site nuclearity and rather than employing a design strategy, they are at the mercy of the way the material crystallizes. This is NOT design since you have no control. This is merely thermodynamics and limited exploration of the current Pd-S system! Further, it does not appear this method provides any availability to access nuclearities other than three.

Our study demonstrates the unprecedented performance of palladium sulfides in liquid-phase alkyne semi-hydrogenation. The observed face-dependency of the geometric and electronic characteristics of the active ensembles, illustrates the concept of catalyst design based on controlling the crystal phase and morphology. However, although we show that the choice of route and support have a strong influence, we realize that we do not yet have the synthetic control to access many different structures. In view of this, **we have now amended the title to ‘Selective ensembles in supported palladium sulfides for alkyne semi-hydrogenation’.**

2. It is unclear how a selective poison ensures facile activation of reactants. One could rationalize selective (one functional group over another) activation, but facile seems counter-intuitive.

We used the term ‘facile’ in reference to the bifunctionality of the Pd₃S(202) surface with respect to the activation of molecular hydrogen and the alkyne. When the sulfur centers are considered, our calculations show that the dissociative adsorption of the hydrogen molecule is more exothermic as hydrogen atoms can be incorporated in the form of thiol groups, leaving the palladium sites free for the adsorption of the alkyne. These conclusions have been experimentally verified by detailed kinetic analyses (please refer to comment #15) demonstrating a bifunctional mechanism with no interference between the alkyne and hydrogen adsorption sites, which contrasts with the competitive mechanism observed on unmodified palladium catalysts. Thus, the presence of sulfur not only acts as a poison (modifying the electronic properties of the ensembles to reduce the adsorption strength of the alkyne and the alkene), but also facilitates the activation of the reactants (ensuring the absence of competition). **We have clarified these points in the manuscript (page 2, line 13).**

3. The DFT calculations suggest the adsorption energy of ethylene on Pd₃S(202) is only 0.08 eV (8 kJ mol⁻¹). This is an exceedingly low adsorption energy. It seems too low! Does ethylene only physisorb to the surface via vdW attractions? Is a di-sigma or pi-bonded species preferred?

The adsorption energy, E_{ads} , of ethylene on Pd₃S(202) was calculated using both the revised Perdew-Becke-Ernzerhof (RPBE) and Perdew-Becke-Ernzerhof (PBE) functionals, yielding values of 0.44 eV (RPBE) and -0.01 eV (PBE), respectively. Upon consideration of the van der Waals contributions, E_{ads} becomes -0.08 and -0.30 eV, which indicates that ethylene physisorbs on the surface *via* these weak attractive forces. Assessment of the H-C-C bond angle in the HCCH intermediate indicates that the molecular fragments adsorb *via* a π -bond on both the Pd(111) and Pd₃S(202) surfaces. **These details are now mentioned in the Supplementary Information (page 24, line 23).**

4. Figure 6a and b needs much more detail than has been provided. Where does this data come from? Are all the adsorption energy difference between acetylene and ethylene calculated by the authors in Figure 6a? How was the difference in adsorption energy (at 0 K) calculated for Pd-HHDMA calculated? How was the polymer HHDMA represented in the DFT calculation? Similar questions of where the data actually came from is apparent for Figure 6b too.

We apologize for the incomplete description of the data presented in this figure (now **Figure 7**). All theoretical parameters (the adsorption energy of the alkyne and alkene, the hydrogen activation energy, the energy of segregation, and the ensemble size) were calculated by us. **Full details of these calculations are now provided in the Supplementary Information (page 15, line 1).** The simulations are at 0 K and the representation of Pd-HHDMA follows our previous calculations (see: *Chem. Eur. J.* **2014**, 20, 5926; *Catal. Sci. Technol.* **2016**, 6, 1621).

5. In the comparison of the catalytic activity, did the authors take any particular measure to assess the role of mass-transfer limitations? There is no discussion of an evaluation of potential mass transfer effects (Madon-Boudart, Koros-Nowack, etc.).

The role of mass-transfer limitations was excluded using the Koros-Nowak diagnostic test. Accordingly, the logarithm of the reaction rate of the catalysts with varying catalyst-to-support ratio was plotted *versus* the logarithm of the amount of active phase (see: *Chem. Eng. Sci.* **1967**, 22, 470). The fitting line features a slope close to 1, which confirms that the reaction proceeded under kinetic control. Also, as reported by Madon and Boudart (see: *Ind. Eng. Chem. Fundam.* **1982**, 21, 438), the same test has been carried out at a higher temperature to exclude the presence of heat-transfer limitations. **These results are now discussed in the manuscript (page 20, line 12) and shown in Supplementary Figure 6 (page 8).**

6. The choice of solvent (benzene and/or toluene) is somewhat puzzling from the standpoint that both of these solvents can be hydrogenated. The experimental temperature are probably too low, but have the authors confirmed there is no competitive hydrogenation? It would be interesting to determine if an ensemble size of Pd₃ is sufficient to hydrogenate arenes.

No trace of fully or partially hydrogenated arene was observed in blank alkyne semi-hydrogenation experiments over the range of conditions investigated. **We have calculated the adsorption of toluene (the solvent) and benzene (the internal standard) and their first hydrogenation.** The adsorption of both solvent molecules on the Pd₃S(202) surface is endothermic (by 0.07 and 0.15 eV, respectively). The first hydrogen addition to both species is endothermic (by 0.64 and 0.63 eV, respectively). The activation energies for this reaction (1.45 and 1.50 eV, respectively) are much larger than the corresponding energy for the alkyne. All evidences confirm the absence of any competitive hydrogenation in our catalytic tests. **These results are now discussed in the manuscript (page 20, line 17).**

7. It is unclear from the description in the experimental section if the turnover frequency (TOF) was calculated based on the total moles of Pd or the surface moles of Pd exposed (i.e., dispersion) in each case.

As now specified in the manuscript (page 20, line 24), the rate was estimated based on the total amount of Pd in a given catalyst.

8. Equation 1 of SI to calculate surface energy doesn't look correct. Why is the surface energy of an unrelaxed surface subtracted from the bulk energy? Can the authors please explain the reason behind this approach for the calculation of surface energy?

8. Regarding the calculation of surface energy, there will be some termination that non-stoichiometric. In the case of non-stoichiometric terminations, the subtraction of the bulk energy seems like a gross averaging of energies of Pd and S and perhaps an over simplification.

We have calculated the surface energy, γ_s , for an asymmetric slab configuration with one side fixed in the optimized bulk geometry and one side fully optimized by the following expression (see: *Nanoscale* **2017**, *9*, 13089):

$$\gamma_s = \frac{1}{2A} [E_{\text{slab}}^{\text{unrelaxed}} - NE_{\text{bulk}}] + \frac{1}{A} [E_{\text{slab}}^{\text{relaxed}} - E_{\text{slab}}^{\text{unrelaxed}}]$$

where A is area of the surface considered, $E_{\text{slab}}^{\text{relaxed}}$ is the energy of the relaxed slab with one side fixed in the optimized bulk geometry, $E_{\text{slab}}^{\text{unrelaxed}}$ is the total energy of the unrelaxed slab prior to surface geometry optimization.

We apologize if the use of asymmetric slabs in the calculations was not fully detailed in the original manuscript. Concerning the presence of non-stoichiometric surfaces, Pd_xS (where x is 3 or 4) materials can be cleaved along directions that generate polar terminations. However, all the observed and predicted surfaces correspond to stoichiometric terminations. In addition, polar reconstructions are unlikely due to the very large covalent nature of the Pd–S bond. **We have now elaborated on the choice of asymmetric slabs in the calculations and on the stoichiometric nature of the relevant surface terminations in the Supplementary Information (page 16, line 9).**

10. The (202) termination of the material is expected to be a significant fraction of the particle surface according to DFT calculations. This facet also has hydrogen “storage” capacity through thiol formation. There has been extensive reporting that “stored” hydrogen in the form of beta-hydrides are more active than ambient hydrogen dissociated on the surface. We recognize thiol H is chemically distinct but has any mechanistic calculations been conducted to identify the propensity of this hydrogen to affect the reaction energy diagram?

The possible formation of (beta-)hydrides was examined carefully. Calculation of the potential contribution of (sub)surface hydrogen species, revealed that the adsorption energy of a hydrogen atom at the surface (–0.03 and –0.48 eV, respectively) or in the bulk (0.14 and –0.18 eV, respectively) are both significantly lower for Pd₃S(202) than for Pd(111). Thus, the simulations on unmodified Pd agree with the experimental observations reported in the seminal study of Teschner *et al.* (see: *Science* **2008**, *320*, 86), whereas the calculations on Pd₃S point towards the absence of hydrides. To corroborate these conclusions, we have studied the temperature-programmed reduction in hydrogen. The analysis on Pd₃S/C₃N₄ did not evidence any peak assignable to beta-hydride formation (e.g. in the temperature range 323–373 K, see: *Catal. Lett.* **2006**, *108*, 159), further discarding any potential role of these species in the reaction mechanism. **The absence of subsurface hydrogen is now discussed in the manuscript (page 9, line 12).**

11. An appropriate and worthwhile addition to this manuscript is a Wulff construction of a Pd₃S particle using the surface energies listed in Table S2.

We have considered the Wulff construction of Pd₃S and in fact the (101) surface, which is related to the observed (202) termination, is predicted to have the highest fraction exposed (see table and figure below). However, **we find it misleading to include these results in the manuscript** since this approach predicts the equilibrium shape of nanoparticles. In particular, the starting point for simulating the Wulff structure (see: *Z. Kristallogr.* **1901**, 449) does not match the synthetic protocol employed to prepare Pd₃S, which involves the transformation of a palladium nanoparticle rather than the growth of the Pd₃S crystal. Indeed, the close geometrical relation observed between the dominant (202) termination of Pd₃S and that of the unmodified Pd(111) precursor, strongly suggests that the sulfidation proceeds under kinetic control *via* the path of least structural distortion.

Total volume and fraction of each of the surfaces for a Pd₃S nanoparticle with a diameter of 2 nm.

Volume / nm ³	Area / nm ²	Fraction exposed					
		(001)	(100)	(010)	(110)	(011)	(101)
49.56	68.23	0.25	0.13	0.14	0.11	0.10	0.27

The rotated views of equilibrium morphology and atomistic representations of Pd₃S nanocrystals predicted by Wulff constructions. The crystal morphology model was created with VESTA (Visualization for Electronic and Structural Analysis) version 3.4.0 package (see: *J. Appl. Crystallogr.* **2011**, 44, 1272).

12. A popular mechanism for oligomerization is the reaction of a partially dissociated acetylene (*i.e.*, vinyl) with acetylene rather than the coupling of two acetylene molecules. What is the motivation for choosing a dimerization reaction pathway between acetylene? The reference by Spanjers *et al.* utilizes a surface reaction between adsorbed acetylene and vinyl to form C₄ products.

We initially focused on the dimerization of acetylene for two reasons. Firstly, as the barrier for the first hydrogenation is large we assumed that alkynes would have a higher surface coverage than vinyl fragments. Secondly, because the choice of the simplest reaction path is justified by scaling relations and other paths can be related to this (see: *ACS Catal.* **2018**, 8, 1662). **We have now complemented the reaction energy diagram for C-C bond formation** by considering both the reaction between acetylene

and a partially-dissociated acetylene molecule, and the reaction between two partially-dissociated acetylene molecules. **The results, including the energy profiles and configurations of the initial, transition, and final states, are presented in Supplementary Figure 21 (page 32).** As anticipated, both reactions have lower barriers for C₄H₅ and C₄H₆ formation on Pd(111) (1.43 and 1.37 eV, respectively) than on Pd₃S(202) (1.56 and 1.85 eV, respectively). These values are also larger than any of the hydrogenation steps, further confirming the absence of oligomer formation during the reaction over the palladium sulfide catalysts under the experimental conditions studied.

13. Regarding the evidence of surface segregation for intermetallic compounds we would recommend the authors cite the HS-LEIS work by Rameshan et al.2-4

We focused on literature reporting the stability of intermetallic compounds in alkyne hydrogenation. However, the works of Rameshan *et al.*, also clearly illustrate the limitations imposed by surface segregation in this type of material. **We have now cited one of the most relevant articles (*J. Catal.* **2012**, 290, 126).**

14. The authors comment on the activity of different catalysts for different alkyne semi-hydrogenations. The presentation of data is incomplete and the authors should comment on the selectivity and product distribution (cis, trans E, Z etc).

The main reaction studied in the original manuscript was the hydrogenation of 2-methyl-3-butyn-2-ol to 2-methyl-3-buten-2-ol (see **Figure 4**). Since this is a terminal alkyne, it does not form double bond isomers. Additional substrates of varying size were also considered to demonstrate the improved active site accessibility of active ensembles in the supported palladium sulfide catalyst compared to the ligand-stabilized palladium nanoparticles, Pd-HHDMA (**Figure 6b**). **As now clarified in the manuscript (page 9, line 8)**, in these reactions the *cis* isomer was preferentially formed over both catalysts, which is attractive due to their relevance in the fine chemical industry. The main side product was the alkane. **To provide a broader comparison with the other state-of-the-art catalysts, we now also compare the performance of all catalysts in the hydrogenation of 1-hexyne.** Only small amounts (<5%) of double bond isomers were detected in any case. **The new data is presented in Supplementary Figure 8 (page 10) and described in the manuscript (page 9, line 3).**

15. The authors make a mechanistic claim regarding the dual site nature of the reaction pathway with regards to alkyne and hydrogen binding. The authors do NOT provide enough experimental evidence to make this statement with absolute certainty. This is a very long-standing debate in the community (at least for acetylene, which the authors use as the model compound for making these assertions through DFT) and surely further experiments and microkinetic modeling must be conducted before the mechanism can be fully established.

To confirm the dual-site nature of the reaction mechanism, **kinetic analyses were comparatively conducted over Pd₃S/C₃N₄ and Pd/C₃N₄** in order to assess the presence or absence of competition between 2-methyl-3-butyn-2-ol and H₂. Interestingly, the reaction order with respect to the alkyne over Pd₃S (0) deviates from that commonly evidenced on Pd nanoparticles (-1). This was rationalized by constructing microkinetic models based on the classical Langmuir-Hinshelwood rate expressions assuming dual or single-site nature, respectively. **The experimental data and derived microkinetic models are now presented in Supplementary Figure 10 and accompanying text (page 11, line 6) and discussed in the manuscript (page 10, line 6).**

16. *The author's claim regarding the resistant nature of these materials to surface segregation is tenuous at best. Though the energetics from DFT support a lack of surface segregation, these calculations are conducted in vacuum. It is important to note that surface segregation is a very strong function of the interface chemical composition (presence of O₂, H₂ etc) and temperature. These conditions are typically not captured through DFT surface calculations. Refer to HS-LEIS work by Stadlmayr et al.⁴ and Kopfle et al.⁵ as examples.*

As correctly pointed out by the Reviewer, surface segregation is strongly dependent on the reaction environment in gas-phase chemistry under relatively high pressures and temperatures. However, in our case reactions are carried out in the liquid phase, meaning that the solvent modulates the effective pressures on the surface by limiting the solubility of H₂ and consequently the induced segregation. To probe the effects of solvation on the adsorption energies of intermediates on the supported palladium sulfide surfaces, we applied the recently developed continuum solvation model into the VASP, a plane-wave based electronic structure code, Multigrid Continuum Model (VASP-MGCM) (see: *J. Chem. Theory Comput.* **2016**, *12*, 1331). **As now reported in Supplementary Table 7 (page 27)**, the solvation energies were found to be very small. Therefore, similar binding energies to those reported for these intermediates in **Supplementary Table 4** are expected in solution.

17. *The section of the manuscript "A long-term test conducted...ensemble control of the surface" does not present a coherent story. We doubt there is any direct one to one correlation between phase separation (presumably the authors are only probing the bulk via XRD?), sintering and compositional changes (once again, the bulk via ICP or STEM?) with the surface composition and ensemble size control. The fact that the material has a stable activity does not imply there is no surface segregation as this phenomenon can occur quite rapidly (at the instant the material is exposed and before the first point is collected).*

It is true that segregation effects can occur rapidly and may be overlooked by bulk characterization techniques. To further evidence the robustness of our supported palladium sulfide catalyst, **we have analyzed the catalyst recovered after 50 h on stream by high-resolution transmission electron microscopy and X-ray photoelectron spectroscopy. The new data, included in Figure 2 (page 33) of the manuscript and in Supplementary Figure 5 (page 6)**, demonstrate the preservation of the crystalline Pd₃S structure and the virtually identical surface electronic states of Pd before and after reaction. This is consistent with the high energy of segregation predicted for this system by DFT (ca. 4 eV) and confirms the high stability of the catalyst under the conditions of liquid-phase alkyne semi-hydrogenation.

18. *Was there any attempt (such as TGA-MS) to quantify C deposition. What does a carbon mass balance in this reaction look like?*

As now clarified in the manuscript (page 20, line 25), no coke deposition was observed in the reaction. The carbon balance, calculated as the ratio between the number of moles of carbon in the products and in the feed, exceeded 98% for every experimental point. Consistently, no difference was noticeable between the thermogravimetric profiles of the fresh and used catalysts. Therefore, we can exclude oligomerization over the Pd₃S/catalysts, which agrees with the higher barriers predicted for oligomerization paths than for any hydrogenation step (see response to comment #12).

19. *We would recommend the authors to cite the relevant paper from Yang et al.⁶*

Thank you for the suggestion. We would like to highlight that a study (see: *Nat. Commun.* **2014**, *5*, 5787) dealing with the in-depth characterization, testing in liquid-phase semi-hydrogenation, and molecular-level understanding of Pd nanoparticles modified interstitially with boron was already cited in the original manuscript (reference 28). **We have now also included the work of Yang et al. (ref. 29).**

20. *Figure 1 is odd and irrelevant. It is more suited for a review paper, which this is not. It is irrelevant because it does not add to the understanding of the molecular origins of the observed activity-selectivity in this system.*

We cannot agree with the Reviewer. Considering the broad readership of Nature Communications and the fast evolution of research in the area during the last five years, we are convinced that a concise overview of alkyne semi-hydrogenation catalysts is critical to put the current research into perspective. We are unaware of any review paper that illustrates the developments in catalyst design for selective hydrogenations of alkyne so clearly. However, we recognize that our efforts to identify important descriptors were not optimally introduced. **During the revision we have paid specific attention to both the balance of providing sufficient molecular understanding on the performance of supported palladium sulfide and the contextualization with other catalytic systems in the Introduction and Discussion.**

21. *The first sentence of the paper makes no sense. A 'heterogeneous catalyst' not 'heterogeneous catalysis' feature a high density of isolated sites.... The first sentence must be rewritten!*

This typo has now been corrected.

22. *What the authors referring to when they believe a PdS surface mimics the "enzymatic strategy of electronic density control"? I am well versed in mechanistic enzymatic catalysis and this statement has little concrete meaning to me. The authors should provide a reference and at a minimum, a description of how PdS approximates electron distribution (?) observed in enzymes.*

With this statement, we referred to the fact that after a long search for enhancing the performance of a very reactive metal for alkyne semi-hydrogenation, the best solution was sulfur, which can be likened to nature's strategy of electronic density control. In particular, this *p*-block element is the primary electronic poison in enzymatic catalyst like nitrogenases and also tunes the electronic levels in Fe₄S₄ clusters found in proteins. **We have now clarified this point in the manuscript (page 4, line 10).**

23. *What is the motivation of choosing 2-methyl-3-butyn-2-ol as a model substrate?*

As now specified in the manuscript (page 8, line 13), 2-methyl-3-butyn-2-ol was selected due to its relevance in the fine chemical industry for the production of Vitamin E (see: *Catal. Today* **2007**, 121, 45; Synthesis of Vitamin E in *Vitamins & Hormones*, **2007**, Academic Press, 155-200; *Angew. Chem. Int. Ed.* **2012**, 51, 12960). To provide a broader comparison among all state-of-the-art systems, we have now conducted and included **temperature and pressure dependence experiments on the reaction rate and selectivity in the semi-hydrogenation of 1-hexyne**, a widely studied model alkyne (see **Supplementary Figure 8 (page 10) and related description in the manuscript on page 9, line 3).**

Reviewer #2

This paper contains information which will be of value to those working in the rational design of solid surfaces for selective catalysis. The work focusses on the use of two sulphur containing structures which both lead to disruption to the Pd ensemble and modify adsorption characteristics of the adsorbates and intermediates. The data for both Pd₃S and Pd₄S is of value and the latter has already been shown to exhibit useful properties in selective hydrogenation as appropriately cited by the authors. However, the origin of the later work was based on an observation that S containing ligands, even after decomposition but with retention of a sulphur adatom/embedded atom, show useful properties in alkyne semi-hydrogenation.

We thank the Reviewer for highlighting the value of our work. To go a step further, a nanostructured Pd₄S catalyst has been prepared following a previously reported recipe by Anderson *et al.* (see: *J. Catal.* **2016**, 340, 10). Interestingly, in contrast to the original calculations, which considered the Pd₄S(110) surface, detailed analysis of the resulting material revealed the prevalence of the thermodynamically most stable (200) termination. This surface was also found to display trimeric ensembles, which are considered responsible for the comparable performance to Pd₃S/C₃N₄ and absence of oligomerization, enriching the results of our study. As mentioned in the Introduction, very few works have studied the modification with *p*-block elements like the sulfur-decorated palladium nanoparticles reported by Anderson *et al.* (see: *J. Catal.* **2011**, 281, 231). Thus, palladium sulfide catalysts could be envisaged as an evolutionary step of the sulfur doping previously reported in terms of improved stability under reaction conditions.

1. As the authors indicate (Fig 1) that they are interested in the historical evolution of these catalysts, this reference (or both) should be mentioned (Chem. Commun., 2011, 47, 2351-2353 and J. Catalysis, 2011, 281, 231–240.

Thank you for the suggestion. **We have included the Pd/TiO₂ catalyst modified with diphenyl sulfide in Figure 1 with appropriate citation (ref. 22).**

2. The reaction work involved 2-methyl-3-butyn-2-ol which is a liquid phase reaction (usually in the presence of solvent) while the calculations are described for a more "simple" 2 carbon alkyne, acetylene (gas phase reaction). I understand why this has been done in terms of computational time/capacity however, the authors should justify why an extrapolation may be performed (we often find that catalysts which show promise for gas phase hydrogenation of e.g. acetylene are not so selective in liquid phase of hex/heptynes and vice versa).

We agree that the use of acetylene as a model compound needs a better justification in the manuscript. In the calculations, the study of a smaller reactive molecule greatly simplifies the search for complex reaction networks or for alternative transition state configurations. This surrogate strategy has limitations, but in general provides robust and fast answers to technically challenging reaction networks. In the present case, to demonstrate the validity of the approach, **key calculations for the adsorption of the 2-methyl-3-butyn-2-ol and the product of hydrogenation are now also presented in Supplementary Figure 17 (page 28)**. Given the parallelism in the adsorption step, and the very similar reaction barriers for the first and second hydrogen additions to 2-methyl-3-butyn-2-ol with respect to acetylene, we can consider that our computational set up is meaningful. **This is now specified in the manuscript (page 9, line 19).**

3. Authors should make clear that Lindlar is usually employed in the presence of quinolone (which I am assuming, is not employed here.

Quinoline was not used in our experiments. Typically, it is only added to Lindlar catalysts with low lead contents (*i.e.*, around 1 wt.% Pb), whereas our lead content (3 wt.% Pb) is at the higher end of the standard range. **This is now specified in the manuscript (page 18, line 18).**

4. p-8. *There is an assumption (not substantiated) that excellent selectivity is a consequence of the absence of Beta hydride in the Pd₃S. Are the authors convinced that no hydride is formed (check refs 11 SI and 51 in main manuscript).*

The possible formation of beta hydrides was also questioned by Reviewer #1 (see comment #10). Indeed, previous studies on Pd₄S/CNF evidenced their formation *via* the temperature-programmed reduction in hydrogen. This was attributed to the solubility of hydrogen in the Pd₄S crystal (see: *Ind. Eng. Chem. Res.* **2007**, *46*, 6313). We have conducted the same analysis on Pd₃S/C₃N₄, but did not observe any peak between $T = 313\text{--}373$ K, which strongly suggests the absence of hydrides. Consistently, DFT calculations show that the incorporation of hydrogen in the bulk is an endothermic process for Pd₃S(202) and Pd₄S(200) (0.14 and 0.25 eV, respectively), supporting the absence of hydride species.

5. *The hydrogen is deemed to be located on the S atoms following dissociation on Pd. Is there any spectroscopic evidence for these thiols? In the DFT work, I was unable to find an energy value given to the subsequent transfer of these hydrogen to the adsorbed intermediates. If hydrogenation or hydrogen transfer take place between a thiol and an adsorbed hydrocarbon, why can't we hydrogenate using a hydroxyl group on an oxide (normally...)?*

In situ diffuse reflectance infrared Fourier transform spectroscopy studies co-feeding C₂H₂ and H₂ over the catalyst was attempted to validate the proposed DFT mechanism. However, it was not possible to detect the formation of thiol groups, suggesting a low concentration on the Pd₃S surface. The strong absorption of the carbon nitride carrier in the same region further reduces the sensitivity of this technique to the intrinsic weak stretching mode of the S–H bond. On the other hand, it is well known that sulfur atoms in transition metal sulfides participate in hydrogen activation *via* thiol formation (see: *J. Catal.* **1992**, *138*, 409; *Catal. Rev.: Sci. Eng.* **2002**, *44*, 651; *J. Am. Chem. Soc.* **2002**, *124*, 7084). Thus, the extrapolation to the Pd₃S system is deemed justified. The involvement of sulfur is also consistent with the distinct kinetic fingerprints evidenced over Pd₃S/C₃N₄ compared to Pd/C₃N₄ (see response to comment #15 of Reviewer #1), **as now explained in the Supplementary Information (page 11, line 6).**

The reason why hydrogenations cannot occur from a hydroxyl group is that this would require the transfer of a proton and also an electron from a reduced metal atom on the surface. This differs from the case of a thiol group, which involves the transfer of a hydrogen atom. The difference arises from the distinct electronic structure of the materials, palladium sulfides are conductive while many oxides are either semiconductors or insulators.

6. *What happens to the sodium which was used as precursor for addition of S? Is any retained? What form does it exist in, where is it located and what consequences does it have in terms of catalytic behaviour?*

Only trace amounts (<2 ppm was quantified by inductively couple plasma optical emission spectrometry) of sodium remain in the catalyst, confirming the effectiveness of the washing step. Thus, we can exclude any role of the cation on the overall activity. **This is now clarified in the manuscript (page 5, line 21).**

7. p-12. *The authors discuss the issue of surface segregation in the context of stability however, given that the highest temperature employed is 363 K, do they believe the barrier to atom diffusion from the bulk is overcome under these conditions (irrespective of what the most thermodynamically stable surface may be)?*

Although the operation temperatures do not exceed 363 K, it is known that the structure of intermetallic compounds can be unstable upon application in the liquid phase under these conditions. For instance, Pd₂Ga was reported to be a very selective and stable catalyst for gas-phase semi-hydrogenation (see: *J. Catal.* **2008**, *258*, 210). However, when the same system was evaluated in the liquid-phase hydrogenation of phenylacetylene, an irreversible oxidative decomposition was observed (see: *J. Catal.* **2014**, *309*, 221). This highlights the importance of considering segregation effects, **and is now stressed in the manuscript (page 14, line 25).**

8. How does (SI, page 15) the calculated heat of adsorption of acetylene compare with the published experimental values?

To the best of our knowledge, there are no published values for Pd₃S, further stressing the novelty of our main catalytic system.

9. The authors measure CO uptakes (Table 1 and SI Table 1)) for samples including Pd₃S yet previous studies (ref 51) show that Pd₄S does not take up (appreciable) amounts of CO. Can the authors add a comment to account for this difference?

As correctly pointed out by the Reviewer, previous infrared studies on Pd₄S did not detect adsorbed CO species. In contrast, we observed CO uptake on both Pd₃S and Pd₄S by pulse chemisorption. This discrepancy could stem from the different experimental setups and protocols applied. Unfortunately no details were given regarding the pretreatment of the catalysts in ref. 54. In our experiments, the catalyst was treated under a diluted H₂ flow to reduce any oxidic species formed upon exposure to air prior to pulsing CO. The calculated adsorption energies, ΔE , indicate that CO should adsorb on all surfaces with the following values: Pd(111), -1.85 eV; Pd₄S(200), -0.57 eV; and Pd₃S(202), -0.65 eV. **These results are now included in Supplementary Figure 25 (page 34) to ensure the reproducibility of the calculations by other researchers.**

Reviewer #3

1. As mentioned by the authors in the manuscript, Pd₄S type of Pd sulfide catalyst has been reported in the literature (e.g. ref. 31), but the direct comparison between the performance of the catalyst developed in this work and Pd₄S is missing. The activity, selectivity and stability of Pd₄S determined experimentally should be included in this work to support the computational results. In addition, it is strongly suggested that the authors state clearly new findings and main difference between this work and those from Anderson's group, otherwise the current work could only be considered as a follow-up work.

We agree that the comparison of Pd₄S for liquid-phase alkyne semi-hydrogenation is relevant to extrapolate the concepts derived for Pd₃S to other palladium sulfide stoichiometry. Reproducing the procedure reported by Anderson *et al.* (see: *J. Catal.* **2016**, *340*, 10), **a catalyst based on Pd₄S nanoparticles supported on carbon nanofibers has been prepared and fully characterized.** Examination of the supported nanocrystals by high-resolution transmission electron microscopy revealed the prevalence of the (100) facet, which matched the computed lowest energy termination. **The results are now presented in Figure 2d (page 33) and Supplementary Figure 2 (page 4).** The catalyst has been extensively evaluated in the semi-hydrogenation of 2-methyl-3-butyn-2-ol and 1-hexyne. **The corresponding results are now discussed in the manuscript (page 8, line 23) and presented in Figure 4 (page 37) and Supplementary Figure 8 (page 10)** and compared to Pd₃S/C₃N₄. Overall, this detailed comparison enriches the conclusions of our work, highlighting the exquisitely balanced geometric and electronic characteristics of ensembles in supported palladium sulfides.

2. I cannot see why the authors tested the catalysts with liquid phase alkynes such as 2-methyl-3-butyn-2-ol, 1-hexyne and 3-hexyne, but did calculations with the simplest alkyne, i.e. acetylene, in gas phase. Although the authors claimed that acetylene “demonstrates a similar adsorption energy and configuration to 2-methyl-3-butyn-2-ol for all the Pd_xS catalysts considered”, it is widely recognized that molecules in the liquid and gas phases are quite different from many aspects, such as the distances between molecules and the structure and entropy of molecules. Therefore, the adsorption free energies of acetylene are totally different from 2-methyl-3-butyn-2-ol under the reaction conditions.

Reviewer #3 is kindly referred to comment #2 of Reviewer #2 for details. Briefly, to validate the use of acetylene for identifying the reaction network, **we have now included key calculations for the adsorption of 2-methyl-3-butyn-2-ol and the hydrogenation product in Supplementary Figure 17 (page 28).** The first and second hydrogenation steps of 2-methyl-3-butyn-2-ol (C₅H₈O) to form 2-methyl-3-buten-2-ol (C₅H₁₀O) on the Pd₃S(202) surface present very similar barriers (within 0.09 eV). Therefore, these results further corroborate the suitability of using the simplest alkyne for unravelling performance differences over the catalysts studied. **This is now discussed in the manuscript (page 9, line 19).**

3. The authors found that the “Pd₃S/C₃N₄ displays a stable performance for 50 h on stream in the selective hydrogenation of 2-methyl-3-butyn-2-ol”. How is the stability of other catalysts the authors used as references, i.e. PdPb/CaCO₃, Pd-HHDMA, and Pd@C₃N₄?

No stability test was performed over the benchmark catalysts, as the robustness of these systems was already demonstrated in the liquid-phase alkyne semi-hydrogenation (see: *Chem. Eur. J.* **2014**, *20*, 5926; *Angew. Chem. Int. Ed.* **2016**, *54*, 11265). **To confirm this under the conditions and in the reaction reported in our contribution, we have now performed a 20 h stability test over PdPb/CaCO₃, Pd-HHDMA, Pd@C₃N₄, and Pd₄S/CNF, which as expected showed no sign of deactivation. These results are now included in Figure 6 (page 40) and Supplementary Figure 9 (page 10).**

4. This point is also related to the stability of the catalysts. The authors included STEM images in Fig.4c to show that Pd₃S displays a stable performance in the hydrogenation of 2-methyl-3-butyn-2-ol. However, they also attributed the high stability of Pd₃S/C₃N₄ catalyst to the higher coupling ethyne-ethyne barrier than Pd, indicating that oligomerization could be suppressed over Pd₃S, the point of which should also be justified experimentally in the manuscript.

This is in line with comment #18 of Reviewer #1. The absence of oligomers was revealed by the carbon balance, calculated as the ratio between the number of moles of carbon in the products and in the feed, which exceeded 98% for all experimental points. This has now also been confirmed by thermogravimetric analysis, which evidenced negligible differences between the fresh and used catalysts. **These points are clarified in the manuscript (page 20, line 25).**

5. According to the equation included in the SI that the authors used for the surface energy calculations, i.e. Eq. 1, it seems that the authors only considered the stability of different terminations of Pd₃S in vacuum. However, they should keep in mind that the catalyst is exposed to hydrogen and alkynes during reaction, and it is highly possible that the surface sulfur is hydrogenated to SH. This will definitely change the surface energies of different terminations obtained by the authors.

This relates to comment #16 of Reviewer #1. In gas-phase applications and under high pressure the surface energy strongly depends on the reactive environment. However, it should be noted that under the liquid phase conditions studied, the solubility of hydrogen is limited and the adsorption energies of hydrogen, the alkyne, and the solvent on the surface are relatively low. For these reasons, the population of reactants on the surface is ultimately unlikely to change the surface energies of the considered materials. This is a major advantage of conducting the reaction in the liquid rather than gas phase where the pressure effects are much more pronounced. **This distinction has been briefly alluded to in the manuscript (page 21, line 13).**

6. It seems that the adsorption of C₂H₂ over Pd₃S is too weak, even weaker than over Au. The transition state of C₂H₂ hydrogenation is even higher than the gaseous state in the energy profile shown in Fig. 5. In addition, while considering the adsorption free energies of H₂ and C₂H₂, one can anticipate that both adsorption are endergonic at the reaction temperature. Have the authors calculated the reaction rates theoretically based on the energies shown in Fig.5a to see whether these energies obtained from DFT calculations support the experimental results that Pd₃S is more active than Pd?

Our calculations indicate that these processes are thermoneutral or slightly endergonic. Since the reaction takes place in an organic solvent, entropic effects would have a significant impact. However, only a few entropic models have been developed to account for these contributions. The adsorption energies are certainly much lower in the liquid phase than in the gas phase, enabling the reaction to proceed under the investigated concentrations and pressures.

The higher activity of Pd₃S compared to Pd can be explained by the findings of the kinetic study. While on Pd both H₂ and the alkyne compete for the same active site (thus the alkyne has a negative reaction order), the mechanism on Pd₃S is bifunctional, i.e., the sites for the adsorption of H₂ (sulfur) and C₂H₂ (palladium) are different. **This is now explained in the manuscript (page 16, line 4).**

7. The usage of $E_{\text{ads}}(\text{C}_2\text{H}_2) - E_{\text{ads}}(\text{C}_2\text{H}_4)$ as a descriptor should be justified.

The difference in adsorption energy between an alkyne and the alkene obtained upon its partial hydrogenation ($E_{\text{ads}}(\text{C}_2\text{H}_2) - E_{\text{ads}}(\text{C}_2\text{H}_4)$) provides a measure of the preferential coverage of the active surface by the alkyne (see: *Catal. Rev.* **2006**, 48, 91; *Front. Chem. Sci. Eng.* **2015**, 9, 142), and hence represents a quantitative indicator of the thermodynamic selectivity. Weaker binding of the alkene also suppresses the possibility of over-hydrogenation. **We have clarified this point in the manuscript (page 12, line 22).**

8. *The activation barrier difference between the formation of H_2CCH_2 and $HCCH_3$ are too small on all surfaces studied. Therefore, zero-point energy corrections and entropy effect should be considered.*

The zero-point energy correction and entropy effect would not change the activation barrier difference between the formation of ethene and ethylidene since both states would be shifted by an equivalent amount in the same direction. **This is now discussed in the manuscript (page 21, line 16).**

Reviewers' comments:

Reviewer #1 (Remarks to the Author):

I have read and reviewed the manuscript, "Selective ensembles control in supported palladium sulfide nanoparticles for alkyne semi-hydrogenation" by Albani for a second time. The paper is improved, but I still have a number of issues with this manuscript. I do not believe this manuscript should be published in Nature Communications, it is far better suited for a topical catalysis journal once the authors develop a correct rate expression from the experimental kinetic data for semi-hydrogenation over Pd₄S (see specific comments below).

1. I have a significant comment that needs to be justified by the authors; I must have missed during my first review but I have significant issue with this paper being published in a Nature journal because of it. It is stated in the Introduction (p. 4) that Pd₃S shares the same active site with Pd₄S, which was also shown to "exhibit promising performance in the gas-phase hydrogenation of simple alkynes impurities in petroleum-derived olefin streams". Why should this paper be published in a Nature journal?!!!! Seems to me either ref. 33, 53 or 54 should have been published in Nature Communications, not this paper!.. The three references appear to be the seminal references, demonstrating the ability of S to modify the active site composition of Pd. Since this work on Pd₄S predates the current submission by a minimum of a year, the description in the abstract of the "unparalleled semi-hydrogenation performance" must be removed. It is simply not true. You have paralled the high semi-hydrogenation performance of Pd₃S!

2. Effective semi-hydrogenation catalysts are exemplified by the example of selective hydrogenation of acetylene in the presence of ethylene. In this example, ethylene is present in many fold excess. For this reaction on Pd, the formation of oligomers (green oil) occurs, but the greater threat to selectivity is the over-hydrogenation of ethylene to ethane. While DFT calculations suggest that ethylene desorption is more facile than hydrogenation to the half-hydrogenated intermediate, I have no doubt that the high(er) temperatures (than DFT calculations), high ethylene concentration and low alkyne concentration (due to conversion). How does the experimental selectivity of your Pd₃S catalyst change with increasing alkyne conversion? High selectivity under these conditions is what makes for a good semi-hydrogenation catalyst and NOT under the conditions of a DFT calculation.

3. The authors state since the reaction is conducted in solvent, the reconstruction/modification of the surface is less likely. Why? This might suggest that the solvent (toluene) binds with intermediate strength to the surface (if it bound too tightly which hard to envision for toluene, it might modify the surface). This is a significant statement with zero proof!

There is an entirely new section in the Supporting Information of the manuscript on kinetic analyses that I have a number of comments on.

4. The plot in Figure S6 demonstrates significant scatter. Remove one point, and the slope of the line is greater (or less) than unity. This has significant bearing on the interpretation of this Madon-Boudart 'like' plot (It is not exactly a Madon-Boudart plot since the space velocity is changing for each data point).

5. There are two mechanisms proposed for alkyne hydrogenation – a single site or dual-site model with the same rate-determining step (addition of first hydrogen to make the half-hydrogenated intermediate). The Langmuir-Hinshelwood rate expressions summarized in the Supplementary information are problematic. They can NOT describe the measured kinetic data (reaction orders in Figure S10). For the single site model, eqn. S4 demonstrates the reaction order in H₂ varies from $-\frac{1}{2}$ to $\frac{1}{2}$; the measured reaction order of 1.5 is well outside this range. Similarly, the reaction order in H₂ from the dual site model (eqn. S7) varies between 0 and $\frac{1}{2}$. Once again the experimentally-determined reaction order is outside the predicted range. These mechanisms are inconsistent with the observed kinetics. The authors incorrectly list single values for the reaction order in alkyne and H₂; these expressions demonstrate the reaction order varies and only obtains a singular value at limiting conditions. Bottom line regarding H₂, the reaction rate expressions can not predict a reaction order of 1.5 in H₂.

6. The reaction order in H₂ is abnormally large on both catalysts. In reality, this might be related to a non-ideal behavior (issue of the rate being proportional to gas-phase H₂ pressure or H₂ concentration (soluble amount in solvent)). The experimental kinetics of semi-hydrogenation over Pd4S need further study.

7. A reaction order as high as 1.5 certainly suggests that under experimental conditions (infinitely displaced from your DFT calculation) suggests that addition of the second hydrogenation is rate-determining. It is highly unlikely the addition of the first hydrogen is rate-determining.

Reviewer #2 (Remarks to the Author):

Authors have dealt with both my comments/concerns/queries and those of the other reviewers to a level which I deem satisfactory. The revised manuscript is acceptable in its present form.

Reviewer #3 (Remarks to the Author):

The authors did address most of the comments, and much more work were added. However, there are still three points I raised awaiting further clarified before the manuscript can be published.

One is related to question 6 I raised. I can see that the authors compared the activities of different Pd-based catalysts for the selective hydrogenation of 2-methyl-3-butyne-2-ol, and the results are shown in Figure 4a,b and Supplementary Fig.7. However, one can see that, although Pd₃S/C₃N₄-423 is more active than Pd@C₃N₄, the activity of Pd₃S and Pd/C₃N₄ are identical. Therefore, it is absolutely incorrect to claim 'the higher activity of Pd₃S compared to Pd'.

I still cannot understand why the authors used $E_{ads}(C_2H_2) - E_{ads}(C_2H_4)$ as a unified descriptor. Which property of the catalysts can be described by this descriptor, activity or selectivity? If it can describe activity, why similar values give rise to quite different rates, as one can see from Figure 7a? If it is a descriptor for selectivity, Figure 7a should be re-plotted to shown what is the exact relation between them.

The authors should be cautious to say that 'The zero-point energy correction and entropy effect would not change the activation barrier difference between the formation of ethene and ethylidene', especially when the energy differences are relatively small. I understand that to include entropy effect is difficult, but I insist the ZPVE corrections should be included in all the energies calculated, otherwise the whole computational part of the work cannot be called the state-of-the-art.

NCOMMS-17-31323A - Response to Reviewers

Comments in *blue* - Replies in black - Actions in **bold**

Indicated figures and page numbers refer to the revised manuscript with changes highlighted

Reviewer #1

I have read and reviewed the manuscript, "Selective ensembles control in supported palladium sulfide nanoparticles for alkyne semi-hydrogenation" by Albani for a second time. The paper is improved, but I still have a number of issues with this manuscript. I do not believe this manuscript should be published in Nature Communications, it is far better suited for a topical catalysis journal once the authors develop a correct rate expression from the experimental kinetic data for semi-hydrogenation over Pd₄S (see specific comments below).

After careful consideration and comparison of the original and current comments of Reviewer #1, we are confused by the highly contradictory nature of his/her report, which we feel do not provides a fair representation of our contribution or the major progress made during the first revision. In addition to developing a novel highly-efficient catalyst (Pd₃S supported on carbon nitride, C₃N₄) for the continuous three-phase semi-hydrogenation of alkynes, a reaction of broad relevance in industrial organic synthesis, for the first time we obtain a detailed molecular-level understanding enabling the derivation of structure-performance relationships over this novel family of materials.

At the valuable suggestion of Reviewer #3, we have compared the performance of our catalyst with the Pd₄S phase reported by Anderson *et al.* (*J. Catal.* **2016**, 340, 10), undertaking the major task not only to reproduce the synthesis, confirm the comparable properties, and conduct extensive catalytic tests, but also to attain a similar level of mechanistic insight into this reference catalyst, aspects which were not addressed in the previous works. This enabled us to confirm the superiority of Pd₃S/C₃N₄ over Pd₄S/CNF for three-phase hydrogenations, and thus to generalize a new concept for the design of selective ensembles in metal-based compounds containing *p*-block elements. We are convinced that the evaluation of palladium sulfides of different stoichiometry also went some way towards addressing the original concerns of Reviewer #1 regarding the lack of design, which we had acknowledged and accordingly moderated the language.

Something that the peer also appears to overlook are the distinctions in catalyst selection for gas- and liquid-phase alkyne hydrogenation. As we have illustrated in **Figure 1**, in gas-phase applications for the purification of olefin streams, the preferred industrial catalyst is PdAg/ α -Al₂O₃, while in the liquid-phase application for the manufacture of fine chemicals, the Lindlar catalyst (PdPb/CaCO₃) has dominated for over 60 years. As clearly evidenced by the different selectivity modifiers and poisons added to the Pd-based catalysts, it is not trivial to transpose a catalyst from one phase to the other and palladium sulfides may comprise one of the exceptional cases. The unique properties are put into perspective by identifying the key criteria for each pillar of catalytic function (activity, selectivity, and stability), from which we are able to determine the hierarchy of the degree of optimization in the principal classes of hydrogenation catalyst. To our best knowledge, such a classification is unprecedented in the literature.

As now also clarified in the reply to comment #5, the original rate expression was correct, but requires the use of the hydrogen partial pressure rather than concentration.

Considering the broad fundamental and practical significance of these achievements, including the development of a sustainable and practical synthesis of palladium sulfide, detailed molecular-level rationalization of the catalytic performance, and the determination of a hierarchy of descriptors impacting on activity, selectivity, and stability, the outcome of our work is unprecedented in the literature and could inspire the broad readership of *Nature Communications* to transpose the developed criteria to other hydrogenations and beyond. Thus, we are convinced that our contribution is fully in line with the scope and thus merits publication in this authoritative journal. **We have revised the introduction and discussion sections of the manuscript to ensure that these points are clearly highlighted.**

1. I have a significant comment that needs to be justified by the authors; I must have missed during my first review but I have significant issue with this paper being published in a Nature journal because of it. It is stated in the Introduction (p. 4) that Pd₃S shares the same active site with Pd₄S, which was also shown to “exhibit promising performance in the gas-phase hydrogenation of simple alkynes impurities in petroleum-derived olefin streams”. Why should this paper be published in a Nature journal?!!!! Seems to me either ref. 33, 53 or 54 should have been published in Nature Communications, not this paper! The three references appear to be the seminal references, demonstrating the ability of S to modify the active site composition of Pd. Since this work on Pd₄S predates the current submission by a minimum of a year, the description in the abstract of the “unparalleled semi-hydrogenation performance” must be removed. It is simply not true. You have paralled the high semi-hydrogenation performance of Pd₃S!

As indicated in the Introduction, it was exactly the promising potential of sulfur and the lack of understanding of its role as a modifier that inspired the scientific concept of our contribution (page 4, line 11). Thus, we are surprised by the confusion of the Reviewer in evaluating the novelty of our work with respect to the previous reports (refs. 33, 53, and 54), which includes the following:

- We report a simple scalable route for the synthesis of a catalyst based on a crystalline palladium sulfide (Pd₃S) with non-equilibrium morphology supported on carbon nitride and demonstrate for the first time its unparalleled functionality in three-phase alkyne semi-hydrogenations.
- We conduct detailed experimental and theoretical analyses to rationalize at the molecular level the ability of sulfur to modify both the geometric leading to well-isolated trimeric ensemble, and electronic properties of the active site, an aspect that had also not previously been addressed.
- Even though the synthetic route to obtain Pd₄S is not sustainable (releasing three moles of toxic gaseous SO₂ or even H₂S for every mole of Pd₄S produced), we study and include the Pd₄S phase upon the suggestion of Reviewer #3. The relevance of extrapolating to another palladium sulfide stoichiometry was also appreciated by Reviewer #2.
- The comparison of two palladium sulfides of different stoichiometry and the observed face dependency of the trimeric ensembles enabled us to generalize the concept of catalyst design based on controlling the crystal phase and morphology to exploit the untapped potential of *p*-block elements in tuning the catalyst properties for the liquid-phase semi-hydrogenation of functionalized alkynes, an aspect that should have pleased Reviewer #1’s original report.

We believe that it is highly unfair to shift the attention to the Pd₄S phase, when its inclusion was a consequence of Reviewer #3 comments, which was not foreseen to be transposable to three-phase semi-hydrogenations. Also, the lack of molecular-level understanding and catalyst design aspects may be the reason for the more specialized nature of the previous literature reports.

2. Effective semi-hydrogenation catalysts are exemplified by the example of selective hydrogenation of acetylene in the presence of ethylene. In this example, ethylene is present in many fold excess. For this reaction on Pd, the formation of oligomers (green oil) occurs, but the greater threat to selectivity is the over-hydrogenation of ethylene to ethane. While DFT calculations suggest that ethylene desorption is more facile than hydrogenation to the half-hydrogenated intermediate, I have no doubt that the high(er) temperatures (than DFT calculations), high ethylene concentration and low alkyne concentration (due to conversion). How does the experimental selectivity of your Pd₃S catalyst change with increasing alkyne conversion? High selectivity under these conditions is what makes for a good semi-hydrogenation catalyst and NOT under the conditions of a DFT calculation.

The impact of the reaction temperature and pressure on the catalytic performance under continuous-flow three-phase operation is shown in **Figure 4**. Over the range of conditions studied ($T = 303\text{-}363\text{ K}$ and $P = 1\text{-}8\text{ bar}$), the alkyne conversion varied from 10-65% and the selectivity levels to 2-methyl-3-buten-2-ol

remained in the interval 95-100%, confirming the robustness of the Pd₃S system with increasing conversion. Due the limitations of our continuous-flow set up, in which it is not possible to reach higher conversion by decreasing the liquid flow rate and increasing the catalyst weight under relevant conditions (*i.e.*, without heavy increase of the total pressure), **we have conducted additional experiments in batch mode over the palladium sulfides and benchmark catalysts**. An impressive selectivity (95%) was preserved even at 80% conversion over the Pd₃S and Pd₄S phases, whereas much more significant losses were evidenced for the Lindlar (selectivity 80%) and Pd-HHDMA (85%) catalysts. **This data is now discussed in the manuscript on page 9 (line 19).**

3. The authors state since the reaction is conducted in solvent, the reconstruction/modification of the surface is less likely. Why? This might suggest that the solvent (toluene) binds with intermediate strength to the surface (if it bound too tightly which hard to envision for toluene, it might modify the surface). This is a significant statement with zero proof!

Surface reconstruction is driven by the high chemical potential of adsorbed species. In the liquid phase, the solvent modulates the effective pressure on the surface, reducing the effective potentials of reactants and products. The binding energy of the solvent is also weak (see **Supplementary Table 8**), and the replacement energies of surface atoms are of lower intensity than those calculated for gas-phase operation where surface reorganization and bulk phase formation are more widely reported. Furthermore, the solvent environment is less aggressive due to the lower temperatures (in this case about 373 K) and pressures typically applied. Provided there is no leaching, the stability of the phases is less compromised and reconstruction phenomena are much less likely to occur under the conditions of three-phase operation. On the other hand, the selective hydrogenation of alkynes in the liquid phase is much more challenging from the point of view of molecular understanding because transport and entropic contributions in the different phases cannot be addressed with simple models with the same degree of accuracy.

4. The plot in Figure S6 demonstrates significant scatter. Remove one point, and the slope of the line is greater (or less) than unity. This has significant bearing on the interpretation of this Madon-Boudart 'like' plot (It is not exactly a Madon-Boudart plot since the space velocity is changing for each data point.).

As specified in the manuscript, we use a Koros-Nowak and not a Madon-Boudart diagnostic test to determine whether the reaction rate is limited by diffusion constraints. The former approach is preferred due to the possibility of keeping the metal dispersion equivalent for each data point by adding the same carrier as a bed diluent, while the latter is based on the preparation of catalysts featuring different loadings, which could eventually cause deviation of the average particle size distribution thus affecting the measured kinetics. To improve the confidence of the experimental observations, **we have conducted additional tests to supplement the data in Supplementary Figure 6 (page 6)**. The new results confirm a slope of 1, demonstrating the absence of mass-transfer limitations, which was our original aim.

5. There are two mechanisms proposed for alkyne hydrogenation – a single site or dual-site model with the same rate-determining step (addition of first hydrogen to make the half-hydrogenated intermediate). The Langmuir-Hinshelwood rate expressions summarized in the Supplementary information are problematic. They can NOT describe the measured kinetic data (reaction orders in Figure S10). For the single site model, eqn. S4 demonstrates the reaction order in H₂ varies from – ½ to ½; the measured reaction order of 1.5 is well outside this range. Similarly, the reaction order in H₂ from the dual site model (eqn. S7) varies between 0 and ½. Once again the experimentally-determined reaction order is outside the predicted range. These mechanisms are inconsistent with the observed kinetics. The authors incorrectly list single values for the reaction order in alkyne and H₂; these expressions demonstrate the reaction order varies and only obtains

a singular value at limiting conditions. Bottom line regarding H₂, the reaction rate expressions cannot predict a reaction order of 1.5 in H₂.

Unfortunately, we realize that we did not properly define the parameters used in Equations 4 and 7. The Reviewer appears to have used the concentration of hydrogen in the liquid (c_{H₂}) to determine the reaction order. In **Equations 4** and **7**, the rate is expressed as a function of c_{H₂}, but experimentally the reaction order is obtained as the derivative of the rate as a function of the gas-phase pressure (p_{H₂}).

$$r = k_4 K_3 \frac{\sqrt{K_2 c_{H_2}}}{\left[1 + K_3 c_{C_5H_8O} + \sqrt{K_2 c_{H_2}}\right]^2} c_{C_5H_8O} \quad \text{Eq. 4}$$

$$r = k_4 K_3 \frac{\sqrt{K_2 c_{H_2}}}{\left[1 + K_3 c_{C_5H_8O}\right] \left[1 + \sqrt{K_2 c_{H_2}}\right]} c_{C_5H_8O} \quad \text{Eq. 7}$$

Thus, in both cases, the concentration needs to be expressed as a function of p_{H₂}, which was already presented in **Equation 2**:

$$c_{H_2} = \frac{sKp_{H_2}}{1 + Kp_{H_2}} \quad \text{Eq. 2}$$

After introducing **Equation 2** in **Equations 4** and **7**, the reaction order with respect to H₂ is calculated as a function of the gas-phase pressure (and not the concentration):

$$n(H_2) = \frac{\partial \ln(r)}{\partial \ln(p_{H_2})}$$

This leads to the values reported in **Supplementary Table 3**. It appears that the differences between gas- and three-phase operation were insufficiently highlighted. **We have added the relative equations used to calculate the reaction orders in Supplementary Table 3 (page 11).**

6. The reaction order in H₂ is abnormally large on both catalysts. In reality, this might be related to a non-ideal behavior (issue of the rate being proportional to gas-phase H₂ pressure or H₂ concentration (soluble amount in solvent). The experimental kinetics of semi-hydrogenation over Pd₄S need further study.

This point is related to comment #5, once the gas-liquid phase equilibria is integrated in the list of equations then the reaction orders for H₂ increased significantly in agreement with the experiments. **We have clarified this in the text to avoid any potential confusion. Kinetic tests have been conducted over Pd₄S and included in Supplementary Figure 10 (page 9).** As expected, for H₂ a similar dependence was observed to the Pd₃S/C₃N₄ and Pd/C₃N₄ catalysts, while for the alkyne Pd₄S/CNF showed a similar behavior to Pd₃S/C₃N₄.

7. A reaction order as high as 1.5 certainly suggests that under experimental conditions (infinitely displaced from your DFT calculation) suggests that addition of the second hydrogenation is rate-determining. It is highly unlikely the addition of the first hydrogen is rate-determining.

The Reviewer is kindly referred to the responses to comments 5 and 6 for clarification of this point. As discussed, transport between the phases is crucial to understand the high reaction order found for H₂. Consistent with earlier studies on alkyne hydrogenation reactions over Pd by Neurock *et al.* (*J. Catal.* **2006**, 242, 1), the first hydrogen addition is found to be the most energy demanding step, and is therefore considered rate determining.

Reviewer #2

Authors have dealt with both my comments/concerns/queries and those of the other reviewers to a level which I deem satisfactory. The revised manuscript is acceptable in its present form.

We thank the Reviewer for recommending the acceptance of our manuscript and for highlighting the high level reached. We are grateful that he/she acknowledges the relevance of the extrapolation of our comprehensive analysis to another palladium sulfide phase.

Reviewer #3

The authors did address most of the comments, and much more work were added. However, there are still three points I raised awaiting further clarified before the manuscript can be published.

The Reviewer is thanked for recognizing the extensive additions to the manuscript. We have carefully considered his/her additional comments to further improve the clarity of the manuscript.

1. One is related to question 6 I raised. I can see that the authors compared the activities of different Pd-based catalysts for the selective hydrogenation of 2-methyl-3-butyn-2-ol, and the results are shown in Figure 4a,b and Supplementary Fig.7. However, one can see that, although Pd₃S/C₃N₄-423 is more active than Pd@C₃N₄, the activity of Pd₃S and Pd/C₃N₄ are identical. Therefore, it is absolutely incorrect to claim 'the higher activity of Pd₃S compared to Pd'.

Thank you for pointing out the incompleteness of this statement. **We have revised the text to specify the higher activity of Pd₃S/C₃N₄ compared to Pd/Al₂O₃ (see page 10, line 20 and page 15, line 13). As now indicated (page 10, line 1),** it is true that Pd/C₃N₄ displays comparable activity to Pd₃S/C₃N₄, which can be attributed to the higher dispersion of palladium on C₃N₄ with respect to Al₂O₃ ($D_{Pd} = 63$ and 37% , respectively).

2. I still cannot understand why the authors used $E_{ads}(C_2H_2) - E_{ads}(C_2H_4)$ as a unified descriptor. Which property of the catalysts can be described by this descriptor, activity or selectivity? If it can describe activity, why similar values give rise to quite different rates, as one can see from Figure 7a? If it is a descriptor for selectivity, Figure 7a should be re-plotted to shown what is the exact relation between them.

A selective alkyne semi-hydrogenation catalyst requires a high stability of adsorbed acetylene and a low stability of adsorbed ethene. Studt *et al.*, (*Science* **2008**, 320, 1320) proposed the use of the CH₃[•] radical as a selectivity descriptor. However, it only accounts for covalent interactions without considering van der Waals contributions, and consequently ethene adsorption was zero or positive for several of the investigated metals. Since then, computational methods have evolved and dispersion contributions have become state-of-the art. This disruptive evolution led to the realization that some of the materials for which alkene adsorption was predicted to be unlikely are actually able to bind them. In view of this, we favor the use of $E_{ads}(C_2H_2) - E_{ads}(C_2H_4)$ as a descriptor in **Figure 7a**, to show how the minimization of this parameter has been widely targeted in all new generations of catalytic systems compared to supported unmodified palladium nanoparticles. **Figure 7a** serves the dual purpose to illustrate that the adsorption energies of the organic fragments are not a good descriptor for the catalyst activity and stability when all the potential families of materials are considered, and therefore we find it a valid and useful comparison to include in the manuscript. Replotting the values against selectivity is not of interest since all of the new-generation catalysts excel in this parameter. The incomplete description provided by this descriptor motivated us to identify a framework and hierarchy of key criteria to describe each pillar of the overall catalyst performance including the catalyst stability (**Figure 7b**).

3. The authors should be cautious to say that 'The zero-point energy correction and entropy effect would not change the activation barrier difference between the formation of ethene and ethylidene', especially when the energy differences are relatively small. I understand that to include entropy effect is difficult, but I insist the ZPVE corrections should be included in all the energies calculated, otherwise the whole computational part of the work cannot be called the state-of-the-art.

We have now calculated the zero-point valence energy (ZPE) contributions, which are included in **Supplementary Table 6 (page 24)**. However, to facilitate literature comparison we prefer to report potential energies in the manuscript due to their more widespread use in heterogeneous catalysis. As previously anticipated, the ZPE correction did not affect the activation barrier difference between the formation of ethene and ethylidene.

Reviewers' Comments:

Reviewer #3 (Remarks to the Author):

The authors have dealt with my concerns and they have done sufficient calculations to improve the paper. I think the manuscript is acceptable, subject to considering the following one issue.

I suggest the authors compare the descriptors used in this work with those already reported in the literature. For example, a descriptor firstly proposed by Yang et al., $E_a - |E_{ad}|$, where E_a and E_{ad} is the hydrogenation barrier and adsorption energy of the alkene, respectively, was found able to predict the selectivity of alkenes formation (see Phys. Chem. Chem. Phys. 2017, 19, 18010-18017, ACS Catal. 2012, 2, 1027-1032 and J. Catal. 2013, 305, 264-276).

NCOMMS-17-31323B – Response to Reviewers

Comments in blue - Replies in black - Actions in **bold**

Reviewer #3

The authors have dealt with my concerns and they have done sufficient calculations to improve the paper. I think the manuscript is acceptable, subject to considering the following one issue.

We thank Reviewer #3 for recommending the acceptance of our manuscript and for highlighting the high level reached.

I suggest the authors compare the descriptors used in this work with those already reported in the literature. For example, a descriptor firstly proposed by Yang et al., $E_a - |E_{ad}|$, where E_a and E_{ad} is the hydrogenation barrier and adsorption energy of the alkene, respectively, was found able to predict the selectivity of alkenes formation (see Phys. Chem. Chem. Phys. 2017, 19, 18010-18017, ACS Catal. 2012, 2, 1027-1032 and J. Catal. 2013, 305, 264-276).

We have extended the discussion on previously reported parameters to include the descriptor proposed by Yang *et al.*, citing relevant literature (reference 65).